# Wind and Phytoplankton Dynamics Drive Seasonal and Short-Term Variability of Suspended Matter in a Tidal Basin

**Gaziza Konyssova[1,2]\*, Vera Sidorenko[1,2], Alexey Androsov[1,2], Sabine Horn[2], Sara Rubinetti[1,3], Ivan Kuznetsov[1], Karen Helen Wiltshire[2,4], Justus van Beusekom[2,5]**

[1] Alfred-Wegener-Institut Helmholtz-Zentrum für Polar- und Meeresforschung, Bremerhaven, Germany
[2] Wadden Sea Station Sylt, Alfred-Wegener-Institut Helmholtz-Zentrum für Polar- und Meeresforschung, List/Sylt, Germany
[3] Dipartimento per lo Sviluppo Sostenibile e la Transizione Ecologica, University of Piemonte Orientale, Vercelli, Italy
[4] Climate Science Trinity College Dublin, Dublin, Ireland
[5] Institute for Carbon Cycles, Helmholtz Centre Hereon, Geesthacht, Germany

**\* Correspondence:**
Gaziza Konyssova
gaziza.konyssova@awi.de
ORCID: 0009-0008-5460-4754

**Abstract**
Suspended particulate matter (SPM) is a key component of coastal ecosystems, modulating light availability, nutrient transport, and food web dynamics. Its variability is driven by a combination of physical and biological processes that interact across temporal and spatial scales. Using the Sylt-Rømø Bight as a natural laboratory and focusing on the period 2000-2019, in this study, we integrate statistical analysis of observational data from the Sylt Roads monitoring program and local meteorological stations, neural network modelling and Lagrangian transport simulations. This multi-method approach enables us to disentangle and quantify the relative roles of tidal and wind forcing, as well as biological processes in shaping SPM concentrations across various time scales, based on near-surface measurements at two monitoring stations.
The findings show that wind intensity dominates short-term SPM variability, particularly at the shallow station, where SPM responds rapidly to local wind-induced resuspension. At the deep station, the wind effects appear with a delay of ~5 days, aligning with tidally induced transport timescales (~133 hours) from shallower resuspension zones, as revealed by Lagrangian simulations. Seasonal patterns are further modulated by both reduced wind intensities and the onset of biological processes, such as phytoplankton blooms, which promote flocculation and subsequent settling in spring and summer. Neural network experiments highlight the shifting seasonal balance between physical and biological controls. The median concentration of SPM decreased by up to 80% from winter to summer. Approximately 40% of this seasonal difference can be attributed to weaker wind conditions, while the remaining ~40% is likely driven by biologically mediated sinking processes.

# 1 Introduction

Suspended particulate matter (SPM) is a key component of coastal systems, influencing a wide range of physical and ecological processes. It consists of a mixture of small solid particles of both organic and inorganic origin suspended in the water column with concentrations, size (Eisma, 1986) and composition varying spatially and temporally (Schartau et al., 2019). The spatiotemporal variation in SPM concentrations is driven by an interplay of hydrodynamic, meteorological, and biological factors, which in turn, regulate nutrient availability, light penetration, and organic matter distribution. This directly impacts ecosystem productivity, including the timing of the phytoplankton bloom in spring (Cadée, 1986) and spatial gradients in primary productivity (Cloern, 1987; Colijn, 1982), and trophic interactions (Dolch and Reise, 2010; Graf and Rosenberg, 1997).

In tidally energetic environments like the Wadden Sea, the suspended matter gradient is kept upright by the density-driven coastward transport of bottom water and tidal straining (Becherer et al., 2016; Burchard et al., 2008; Flöser et al., 2011). Sediment accumulation is further influenced by hydrodynamic retention mechanisms such as enhanced settling due to the landward dissipation of current velocity, the scour-lag (Dyer, 1995; Friedrichs and Aubrey, 1988) and settling lag effects (Postma, 1967), and the tidal asymmetry formed by the presence of non-linear processes within the tidal system (Dronkers, 1986; Fofonova et al., 2019; Friedrichs and Aubrey, 1988; Hagen et al., 2022). The role of tidal currents in the variability of SPM concentrations is found to be significant through transport processes (Bartholomä et al., 2009; Christiansen et al., 2006) as well as sediment erosion even under calm and moderate weather conditions (Bartholomä et al., 2009; Lettmann et al., 2009). While tides regulate advection and maintain a baseline shear stress, the co-occurrence of strong winds and wave-induced turbulence during severe weather further enhances bottom stress and promotes additional erosion.

Wind stress and wave action are critical in short-term SPM dynamics. Wind-induced resuspension, particularly in shallow coastal areas, causes episodic increases in turbidity (Aarup, 2002; Fettweis et al., 2012). Stronger and more persistent wind forcing during winter maintains higher SPM concentrations, keeping fine sediments in suspension (de Jonge and van Beusekom, 1995; van Beusekom et al., 1999), while calmer conditions in summer enable enhanced settling (Bale et al., 1985; Verney et al., 2009). Beyond physical resuspension and transport mechanisms, biochemical processes also influence SPM concentrations by modulating aggregation, stabilization, and vertical flux of particulate matter (de Jonge and van Beusekom, 1995; van Beusekom and de Jonge, 2002; Verney et al., 2009). For example, flocculation, the aggregation and breakup of particles, is a key mechanism by which biological activity modulates SPM concentrations (Wotton, 2004; Eisma, 1986). Phytoplankton blooms in spring and summer can promote flocculation, leading to enhanced particle settling and reduced SPM concentrations in the water column (de Jonge & van Beusekom, 1995; Schartau et al., 2019). The process is particularly enhanced by the occurrence of extracellular polymeric substances (EPS), including marine gels such as transparent exopolymeric particles (TEPs), but can also occur due to cohesive properties of fine-grained minerals like clays (Passow, 2002; Verney et al., 2009). The size and cohesiveness of these biologically mediated flocs also govern their settling velocities and resuspension thresholds, affecting how quickly particles are redistributed in the water column. Warmer temperatures also accelerate the decomposition of organic matter (OM), which is nearly always present

in natural flocs in various forms (detritus, adsorbed OM molecules, and living OM), and may range from a minor to a dominant fraction of total floc mass (Eisma, 1986; Engel and Schartau, 1999). This directly links SPM dynamics to food-web functioning (Wotton, 2004; Engel & Schartau, 1999). In addition, temperature also affects top-down controls such as zooplankton grazing, which alters SPM composition and concentration through multiple pathways. Grazing modifies the SPM field not only through phytoplankton consumption but also via the fragmentation of aggregates, production of fecal pellets and organic material that promote microbial and TEP-mediated aggregation. The effects of these processes can either promote particle settling or enhance recycling and suspension, depending on community composition, feeding modes, and environmental conditions (e.g., Passow, 2002; Toullec et al., 2019; Turner, 2002).

In the shallow coastal systems, light can reach the seafloor, stimulating benthic algae growth, provided water clarity allows sufficient light penetration (Loebl et al., 2007). These benthic processes further influence SPM concentrations through both stabilization and removal mechanisms. While microphytobenthos, consisting of benthic diatoms and cyanobacteria, produce biofilms that stabilize sediments and reduce resuspension (Stal, 2010), filter-feeding organisms, such as mussels (*Mytilus edulis*) and oysters (*Magallana gigas*), alter SPM dynamics by removing fine particles from suspension, affecting both sediment deposition rates and nutrient cycling (Graf and Rosenberg, 1997). Moreover, excessive nutrient loads can enhance phytoplankton blooms, whose decay products and extracellular polymeric substances facilitate biomineral floc formation (Passow, 2002; van Beusekom et al., 1999), which promotes OM sedimentation and alters benthic-pelagic coupling by influencing the availability of organic matter to both suspension and deposit feeders. These processes collectively shape coastal sediment dynamics, contributing to the long-term evolution of tidal flat environments.

While hydrodynamic and wind-driven influences on SPM have been extensively studied, their interaction with biological processes and relative contributions remain incompletely understood, despite growing interest. By leveraging long-term ecological monitoring data from the Sylt Roads program, this study aims to quantify the contributions of tidally induced and wind-driven resuspension and transport, as well as biologically mediated processes, to the spatiotemporal variability of SPM in the Sylt-Rømø Bight.

This analysis integrates high-resolution in situ measurements from two Long-Term Ecological Research (LTER) stations from 2000 to 2019, meteorological data from the station List (Sylt, Germany), and outputs from hydrodynamic and neural network modelling to evaluate SPM dynamics across short-term (hourly to daily) and seasonal timescales. The following research questions guide the study: (1) What are the dominant mechanisms driving SPM variability across different temporal scales? (2) How do these mechanisms differ between the two monitoring stations within the Sylt-Rømø Bight? (3) How important are biological processes in shaping observed SPM concentrations?

## 2 Data and Methods

### 2.1 Area Description

The investigations were carried out in the Sylt-Rømø Bight, a shallow, tidal basin in the northern Wadden Sea (southeastern North Sea; see Fig. 1). The basin is semi-enclosed due to two causeways at its northern and southern ends, isolating it from neighboring basins. Its only connection to the North Sea is through the deep tidal inlet Lister Deep, between the islands of Sylt and Rømø. The bay spans approximately 410 km² and features a highly variable topography, including extensive intertidal flats (>45%), shallow subtidal zones (~35%), and deep tidal channels (~10%). The Sylt-Rømø Bight's bathymetry is characterised by a mean water depth of approximately 4 m, with a maximum depth of about 37 m observed in the tidal inlet Lister Deep. Most subtidal and ~72% of intertidal sediments are sand-dominated. The basin also features a small sheltered embayment Königshafen, with an average depth of ~2m and encompasses large areas which become exposed at low tides.

The tidal range in the bight averages 2 m, based on observations at the List tide gauge (E.U. Copernicus Marine Service Information, doi.org/10.48670/moi-00036). Tidal forcing accounts for over 80% of depth-averaged velocity variability under regular wind conditions, in the absence of storms, and over 90% during spring tides (Fofonova et al., 2019). The bight receives minimal fluvial input, with small rivers such as the Vidå and Brede Å (see Fig.1), contributing only 4–10 m³/s of freshwater (Purkiani et al., 2015). Water exchange with the open North Sea occurs exclusively through Lister Deep, a 2.8 km wide tidal inlet. At the mouth of Lister Deep, a prominent ebb-tidal delta extends seaward, acting as both a sediment trap and a pathway for sediment redistribution within the bight (Dissanayake et al., 2012). Localized anthropogenic disturbance may also influence sediment availability and distribution in the region, although detailed records of maritime traffic, dredging operations, and benthic trawling specific to the Sylt-Rømø Bight are limited. However, studies from the broader Wadden Sea show that anthropogenic activities such as bottom trawling (e.g., Bruns et al., 2023; Depestele et al., 2016), dredging (de Jonge and de Jong, 2002; de Jonge, 1983; van Maren et al., 2015), and mussel farming (Jansen et al., 2023) are recognized drivers of sediment disturbance and resuspension.

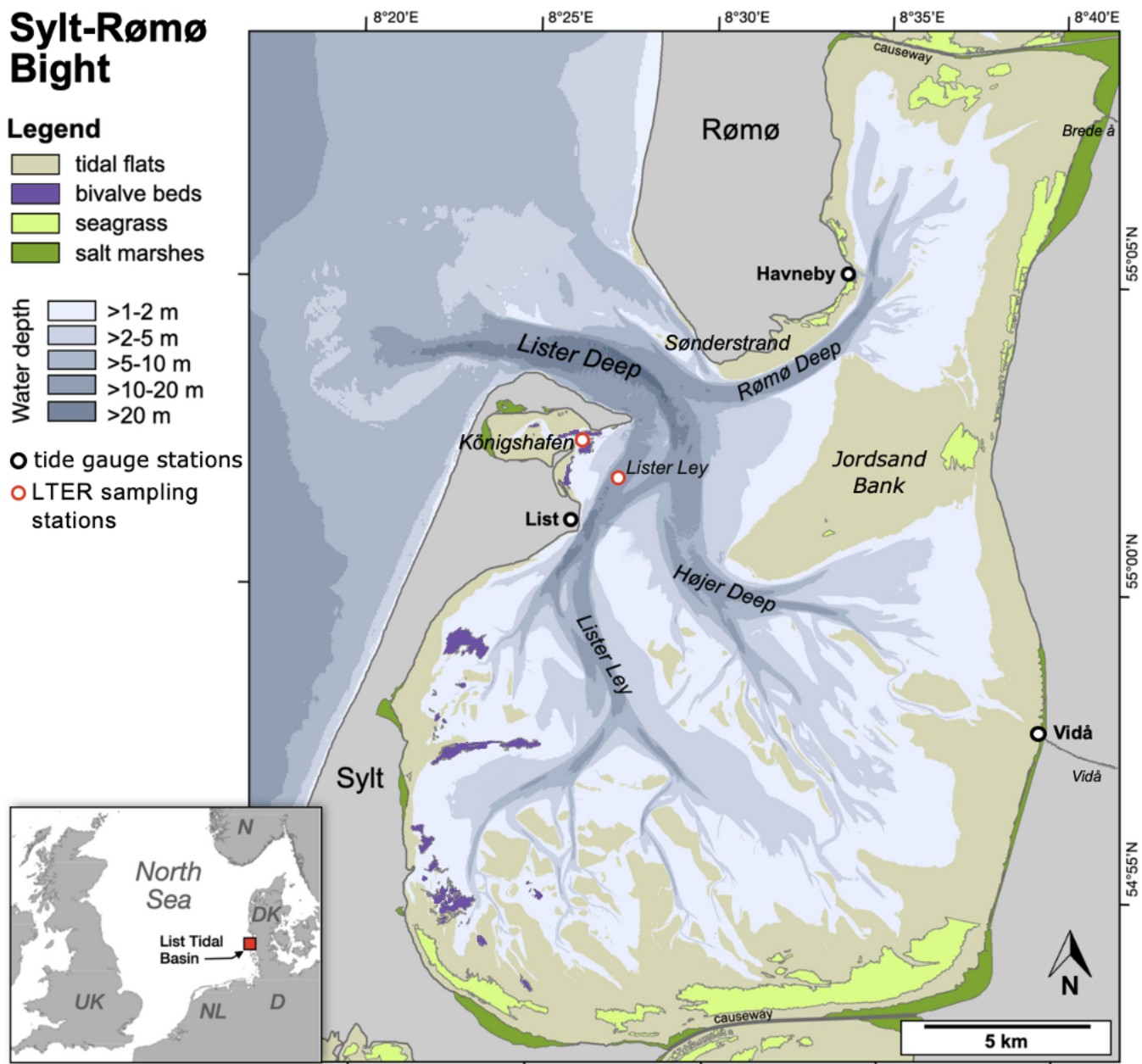

Figure 1: Map of the Sylt-Rømø Bight showing bathymetry, key habitats (tidal flats, bivalve beds, seagrass meadows, and salt marshes), LTER sampling stations (Deep Station – Lister Ley; Shallow Station – Königshafen), tide gauge locations (List, Havneby, Vidå), and rivers (Brede Å and Vidå). The basin is connected to the North Sea via the tidal inlet Lister Deep and laterally enclosed by causeways to the north and south.

## 2.2 FESOM-C Model

This study used the coastal hydrodynamic model FESOM-C (Androsov et al., 2019). FESOM-C is designed explicitly for high-resolution coastal applications and employs a finite-volume cell-vertex

discretization on unstructured meshes composed of triangles and quadrilaterals. This allows for flexible
spatial resolution down to several metres, suitable for simulating complex coastal dynamics (Fofonova
et al., 2019; Kuznetsov et al., 2020, 2024; Neder et al., 2022; Sprong et al., 2020; Sidorenko et al.,
149 2025).

## 2.2.1 Model Setup

The setup utilized an unstructured hybrid mesh of 208,345 nodes and 211,545 elements. Due to the
semi-enclosed state of the bight, the mesh contains a single open boundary at the seaward edge of the
domain, connecting the basin with the North Sea. The horizontal spatial resolution varies from up to 2
m in wetting-drying zones to 304 m in the deeper outer part (near the open boundary). The experiments
were carried out by running 2D barotropic simulations with the wetting/drying option enabled to
capture the periodic submergence and exposure of intertidal areas. The model timestep was set to ~ 0.25
seconds, with data output every ~20 minutes of simulation time. The bottom friction coefficient was
applied as 0.0025, a value identified as optimal in prior studies of the same study area when using
TPXO9 tidal solution (Fofonova et al., 2019; Konyssova et al., 2025).
The simulations are driven by tidal forcing alone, applied at the open boundary. For an accurate
simulation of the tidal dynamics, thirteen major tidal harmonic constituents (M2, S2, N2, K2, K1, O1,
P1, Q1, Mm, Mf, MN4, 2N, and S1) and two over-harmonics (M4, MS4) were prescribed by their
phases and amplitudes at the open boundary based on TPXO9 tidal atlas (Egbert and Erofeeva, 2002).
This selection of the tidal solution was justified by its robust performance and is one of the most
optimal for the North Sea (Fofonova et al., 2019). The current setup has been validated in a previous
work by Konyssova et al. (2025). The model's performance has been validated using tidal gauge (TG)
data from stations List, Vidå, and Havneby (see Fig. 1; performance results are provided in the
Supplementary Materials, Table S1). Since the numerical setup remains unchanged, we refer to
Konyssova et al. (2025) for full validation details.

## 2.2.2 Lagrangian Module

To assess tidally driven spatial connectivity and transport timescales within the basin, we performed
Lagrangian simulations using FESOM-C Drift, a post-processing tool designed for particle tracking.
The model advects massless passive particles based on the velocity fields produced by the
hydrodynamic model. In this study, we use the term "passive tracers" to denote Lagrangian particles
without weight or settling properties. This is equivalent to virtual Lagrangian particles commonly
applied in particle tracking studies and should not be confused with dye experiments that represent
concentration changes in space and time.
The experiment involved releasing passive tracers from all grid elements within the domain that are
consistently inundated during every flood phase. We released about 90,000 tracers at three-hour
intervals over six weeks (169 iterations in total). Each tracer was tracked for up to three weeks and was
removed from the simulation once it reached either of the two Sylt Roads sampling stations (see Fig. 1
and Section 2.3.1). The iterative release process was designed to capture the full range of tidal
conditions and the complexity of hydrodynamic transport within the basin, ensuring that the results are
statistically robust. Upon arrival at a station, tracers were immediately removed to prevent post-arrival
movements from influencing the mapped source regions and transport pathways. The simulations were
conducted independently for each station, focusing exclusively on the paths from the release locations
to the respective sampling site. If a tracer did not reach the designated station within the simulation time
frame, it was considered to originate from a region that falls outside the station's dominant transport
pathways.
The first analysis approach assessed the source regions of passive tracers arriving at the sampling
stations. This allows us to evaluate the connectivity between different subareas of the basin and the
sampling sites. The probability of SPM originating from a given area was mapped based on the
cumulative occurrence of tracer pathways across all iterations. Higher probability values indicate areas
that more frequently serve as source regions or transport pathways for SPM reaching the sampling
stations.
The second part of the analysis was conducted to estimate the mean transit time of the tracers reaching
the sampling stations from shallow source zones, where resuspension typically occurs (defined here as
areas <2 m deep, based on de Jonge & van Beusekom, 1995). The mean transit time over all
implementations was calculated for all elements whose tracers reached the stations within the simulated
three weeks. To quantify how long it typically takes for high tracer concentrations to reach the station,
we computed a probability-weighted median transit time, where mean transit times were weighted by
their probability values. This approach ensures that frequent transport pathways are given greater
influence in the median transit time calculation, reducing bias from rare, low-probability trajectories.
**2.3 Data**
**2.3.1 Biogeochemistry data**
This study used data from the Sylt Roads long-term ecological monitoring program, focusing on a
subset from 2000 to 2019 to ensure consistent methodology and regular sampling. From a broad range
of hydrographic and biogeochemical parameters covered in the dataset, this study specifically analyzed
suspended particulate matter (SPM; mg/L, filtered through 0.4 μm nucleopore filters, rinsed with
distilled water, stored frozen, and dried at 60 °C) and chlorophyll-a (Chl-a; μg/L, filtered through GF/C
filters (Whatman), stored at −20 °C, and extracted using 90% acetone and analyzed
spectrophotometrically using the trichromatic method as described by Jeffrey and Humphrey, (1975).
Both parameters were measured twice weekly at a sampling depth of 1 m below the surface at two
primary stations: the deep station at the Lister Ley channel and the shallow station at the entrance of
Königshafen embayment (see Fig. 1). The full dataset is publicly available on the data portal
PANGAEA (see Data Availability section for DOIs), and the recent evaluation is detailed in Rick et al.
217 (2023).
To analyze seasonal variability, we defined the seasons based on observed cycles in Chl-a
concentrations at the study site, rather than calendar or astronomical definitions. Specifically, we used:
- winter (November 20 – February 19, low biological activity, low Chl-a);
- spring (February 20 – May 31, phytoplankton bloom initiation and peak);
- summer (June 1 – September 19, post-bloom conditions, high light, reduced Chl-a);
- autumn (September 20 – November 20, transitional period).

## 2.3.2 Meteorological data

For the statistical analysis, we also downloaded the quality checked historical meteorological data for
station 3032, List auf Sylt, from Climate Data Center (CDC) of the Deutscher Wetterdienst (DWD).
The data includes hourly mean wind speed and wind direction (dataset ID: urn:x-
wmo:md:de.dwd.cdc::obsgermany-climate-hourly-wind), and daily sunshine duration (dataset ID:
urn:wmo:md:de-dwd-cdc:obsgermany-climate-daily-kl).

## 2.3.3 Sea Surface Height data

Using the validated model setup (Fofonova et al., 2019; Konyssova et al., 2025), the sea surface height
(SSH) data were reconstructed for the deep and shallow stations. In particular, the amplitudes and
phases of the 15 harmonics mentioned above were obtained from the modeling output using Fast
Fourier Transform analysis. Subsequently, the SSH  signal was reconstructed for the exact timestamps
of the LTER sampling using the *t_tide* package (Pawlowicz et al., 2002). This allowed us to estimate
the tidal elevation and phase (through the SSH and SSH gradients) at the time of sampling, which were
then included as input features in the statistical and neural network (NN) analyses. Their relationship is
illustrated and discussed in Supplementary Material, Figs. S3–S4.
**Table 1. The physical and hydrochemical parameters used in the study (short name, units, frequency, and source)**

| Data | Unit | Frequency | Source |
|---|---|---|---|
| SPM | mg/L | twice weekly | Sylt Roads Marine Observatory |
| Chl-a | µg/L | twice weekly | Sylt Roads Marine Observatory |
| Wind speed and direction | m/s, degrees | hourly mean | Deutscher Wetterdienst |
| Light | hours | daily | Deutscher Wetterdienst |
| SSH | m | every 5 minutes | Model reconstruction using 13 harmonics and 2 over-harmonics from TPXO9 and verified with the tidal gauges |

## 2.4 Neural Network

To assess the relative contribution of biotic conditions to SPM concentrations, we employed a forward
NN to predict SPM based on environmental input parameters. The approach relies primarily on long-
term observational data, thereby enhancing the robustness of the results, complemented by model-
derived SSH and its temporal gradient data. Although numerical simulations are also powerful tools in
this context, any discrepancies they produce can largely be attributed to the choice of numerical
methods and the associated uncertainties, including unresolved processes, boundary conditions, and the
spatial and temporal resolution of both modelled and observed data (Skogen et al, 2021). While it does
not provide process-based understanding, the NN approach is particularly well-suited for this task, as it

offers a cost-efficient and data-driven means to capture complex, non-linear relationships between influencing factors. We conducted several sensitivity experiments (not shown), varying both the network depth and the number of neurons per layer. For the current application, increasing the network depth further did not improve performance.

For the first part of the experiment, the primary regression task involved predicting SPM concentrations during the winter season, which is characterized by low biological activity, as indicated by minimal Chl-a concentrations. Focusing on winter allows for a clearer assessment of physical (abiotic) drivers, such as wind forcing, with reduced biological confounding. The input features, 19 in total, of the NN model include wind magnitude at the time of sampling and averages over a series of prior time intervals (6 to 240 hours, corresponding to the intervals analyzed in Subsection 3.2.2), dominant wind direction over 6 and 12 hours (even though the correlation analysis shows only a minor impact, the non-linear effects of wind direction may still be present), salinity, SSH, and the temporal gradient of SSH (computed using a forward scheme). The results of the winter model are presented in Section 3.3.1. To extend the analysis across seasons, we applied the same model architecture to the full dataset, initially using the same input features as in the winter setup (results in Section 3.3.3), and subsequently incorporating two additional features: temperature and the weekly sum of sunshine hours before the measurement date (21 features in total, Section 3.3.4). The latter serve as a pragmatic proxy for both Chl-a concentration and benthic algae abundance. While we do not attempt to separate these mechanisms individually, this approach is exploratory and intended to approximate their combined influence on seasonal SPM variability.

The technical details of the NN architecture are provided in the Supplementary Material, together with the complete predictor list in Table S2.

## 3 Results

Figure 2 presents the time series of SPM concentrations at the deep and shallow stations in the Sylt-Rømø Bight from 2000 to 2019, based on data from the Sylt Roads monitoring program (Section 2.3.1; station locations shown in Fig. 1). Both stations display a pronounced seasonal cycle, with SPM concentrations typically peaking in winter and declining during summer. The deep station shows more frequent and sustained seasonal peaks in SPM concentrations throughout the time series, whereas the shallow station tends to exhibit higher concentrations during peak events. It is also important to note that the regular sampling at the shallow station was discontinued after 2013 following the replacement of the research vessel, resulting in reduced data coverage in subsequent years.

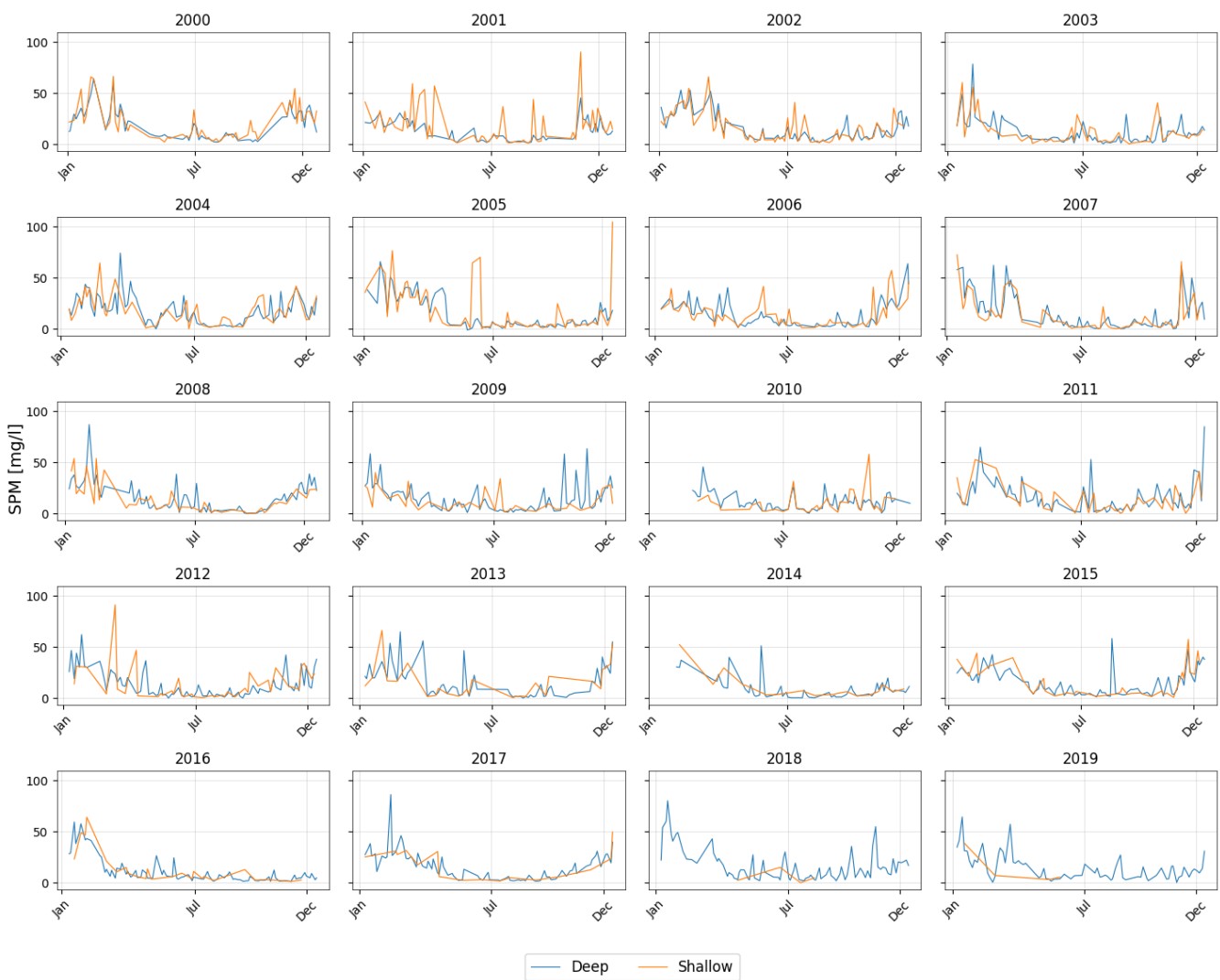

Figure 2: Time series of SPM for the considered years (2000–2019) for the deep (blue) and shallow stations (orange) gridded per year. Each subplot represents a calendar year with time on the x-axis and SPM concentration [mg/l] on the y-axis.

## 3.1 Seasonality of SPM concentrations

SPM concentrations show an apparent seasonality, which is further investigated in relation to biological activity and meteorological drivers in the subsequent analyses. Using Chl-a as a proxy for phytoplankton biomass, this subsection applies statistical analysis to examine how biological activity and wind forcing together shape seasonal patterns of SPM variability at both the deep and shallow stations.

### 3.1.1 Role of biological processes in seasonal SPM variations

Both stations exhibit distinct seasonal variations in SPM and Chl-a concentrations (Fig. 3). SPM concentrations are highest in late autumn and winter, with median values of about ~28 mg/L (33.9 ± 18.2 mg/L) at the deep and median of ~26 mg/L (38.6 ± 16.4 mg/L) at the shallow stations. In January, the peak reaches over 60 mg/L at both stations, with individual events exceeding 70 mg/L. The decline in concentrations is observed from February to May, reaching their lowest values in June to August (around 2-3 mg/L), although occasional peaks above 30-40 mg/L still occur. From September onward, SPM begins to increase. This pattern is similar at both stations, though the shallow station generally has slightly higher SPM concentrations, suggesting potential differences in sediment availability or resuspension dynamics.

Chlorophyll-a concentrations follow an inverse seasonal pattern. It remains low in December–January with a median around 2 µg/L (2.1 ± 0.7 µg/L) at both stations. It increases sharply in early spring, with peaks in March-April exceeding 30 µg/L at the deep station and 25 µg/L at the shallow station. Concentrations then drop in summer, followed by a secondary increase in August–October, when values up to ~15-20 µg/L are observed.

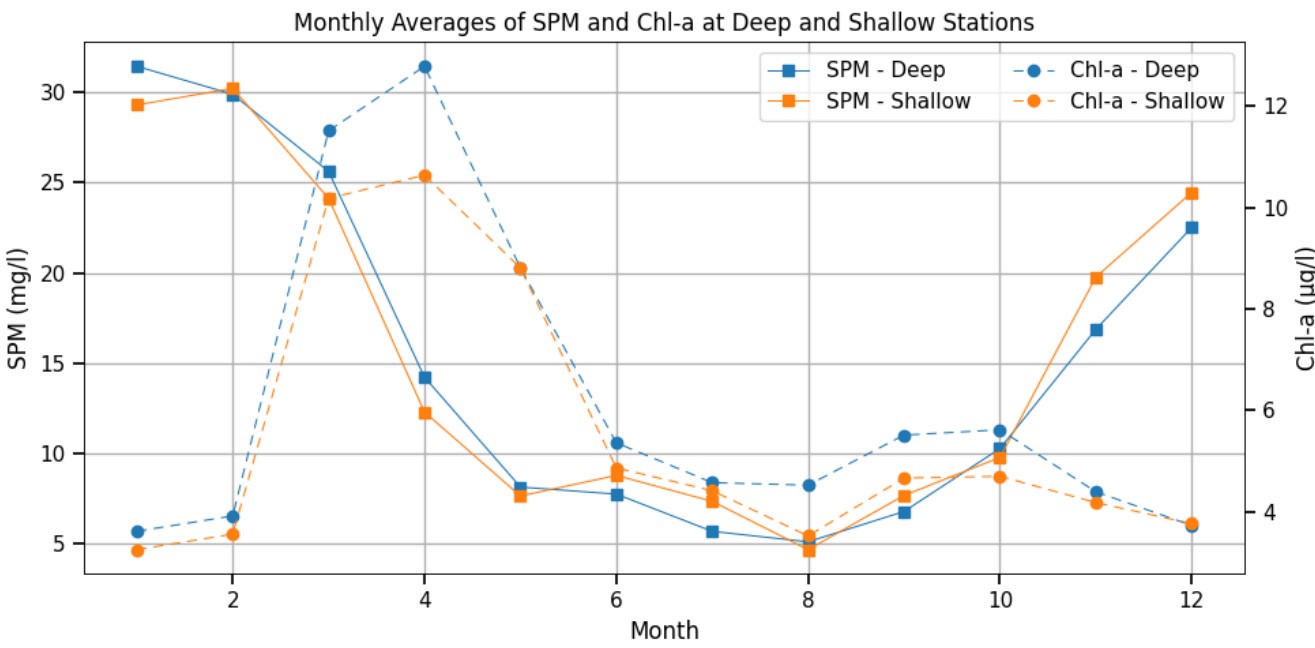

**Figure 3: Monthly averages of SPM and Chl-a at the deep (blue) and shallow (orange) stations for the considered years (2000–2019). SPM concentrations  solid line with square marker, left axis) and Chl-a concentrations (dashed line with circle markers, right axis) are displayed separately for clarity. The x-axis represents the months from January to December.**

The relationship between Chl-a and SPM varies across seasons (Fig. 4). During December–February, biological activity is low and Chl-a constitutes a relatively constant fraction of total SPM, both subject to the same resuspension-deposition processes. This co-settling behavior leads to strong positive correlations observed at both stations. The highest values occur at the shallow station, with $R^2 = 0.84$ in December and $R^2 = 0.75$ in January. At the deep station, correlations are relatively lower, reaching $R^2 =$

0.52 in December and $R^2$ = 0.56 in January. The strong correlation and the slope are in line with the
resuspension of microphytobenthos as observed in winter by de Jonge and van Beusekom (1995). From
March onward, the relationship becomes more complex. As Chl-a rises rapidly during the spring bloom
and SPM concentrations decline, the correlation at both stations weakens with $R^2$ values dropping to
near-zero in March. This divergence reflects differences in vertical distribution and settling behavior.
While phytoplankton is a component of SPM and both are retained on the filter, their decoupling in
spring and summer may reflect that not all phytoplankton is floc-associated. Active phytoplankton may
remain suspended near the surface, while denser biomineral flocs sink more rapidly, leading to weaker
correlations in integrated surface samples. This non-linear and temporally variable influence of Chl-a
also underlies the decision not to use it directly as a predictor in the neural network (Section 3.3), but
instead to approximate biological activity through more general proxy variables such as temperature and
light availbility (see Table S2 in Supplementary Material). During June–August, as both the SPM and
Chl-a concentrations reach low values, the correlation weakens further, with August showing almost no
correlation at the shallow station. In November and December the correlation strengthens again as the
role of biological activity reduces and physical drivers become dominant again.

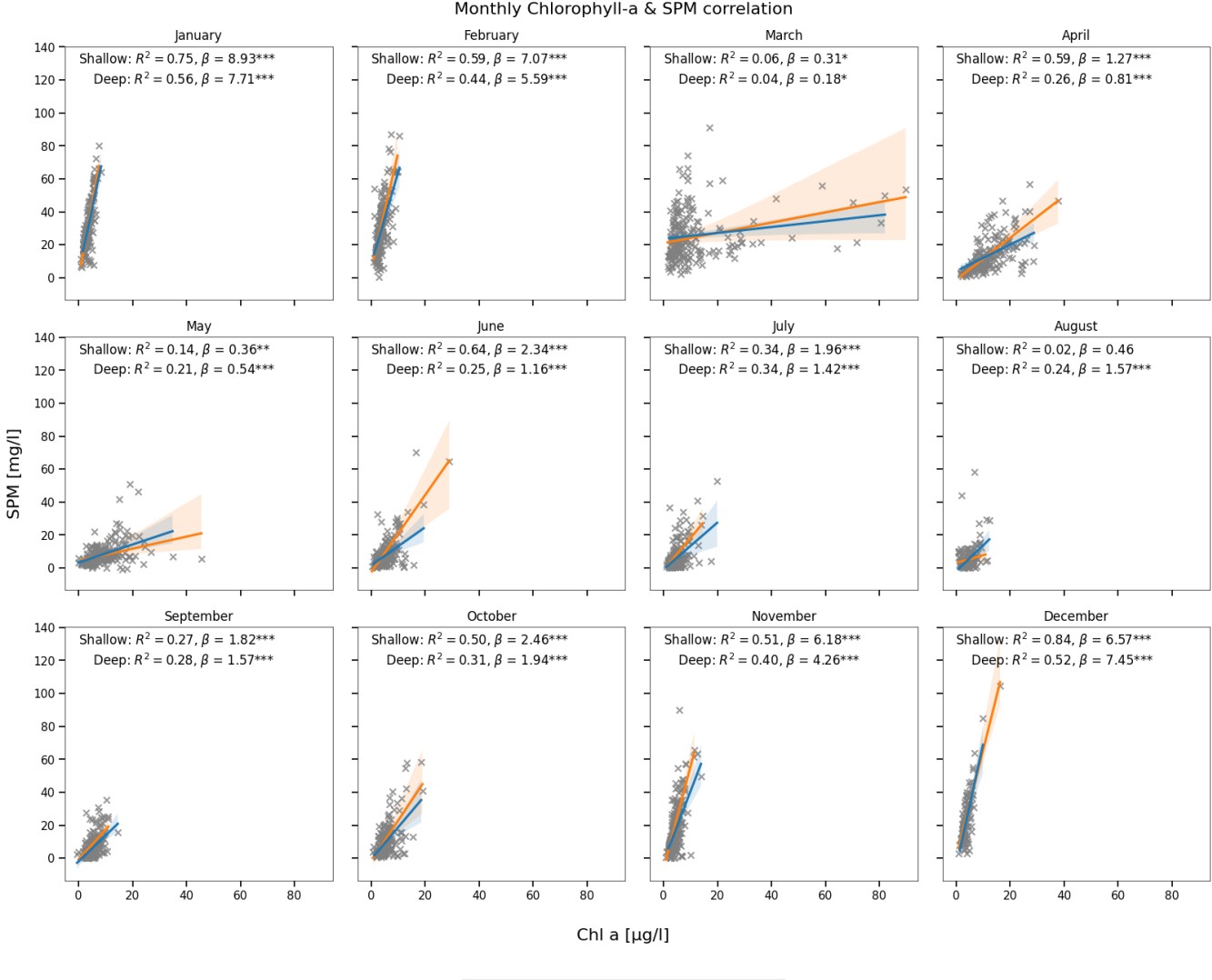

**Figure 4: Monthly correlation between Chl-a (x-axis) and SPM (y-axis) concentrations at the deep (blue) and shallow (orange)**
**stations. The coefficient of determination (R²), slope (β), and statistical significance (\*\*\*p < 0.001, \*\*p < 0.01, \*p < 0.05) are**
**indicated in each panel.**
## 3.1.2 Role of wind forcing in seasonal SPM variations
The relationship between seasonal SPM concentrations and wind characteristics is illustrated in Fig. 5.
Wind speeds are highest in January, averaging around $8.5 \pm 2.8$ m/s, with peaks reaching $14.2 \pm 1.7$
m/s, and gradually decline through spring months, reaching a minimum with mean speeds around $6.5 \pm$
$3.1$ m/s and minima of $2.0 \pm 0.7$ m/s in July–August. From August onward, wind speeds begin to rise
again. This seasonal cycle aligns with the observed variability in SPM, with higher concentrations in
winter and lower values in summer. Dominant wind directions also vary throughout the year, with
westerly and northwesterly winds prevailing in most months, easterly winds in April, and more
southerly winds during October–November. Seasonal wind roses illustrating these patterns are shown in
the Supplementary Material (Fig. S1). However, no clear linear relationship emerges between dominant
wind direction and seasonal SPM concentrations. To account for potential non-linear interactions that
may still play a role, wind direction was included as an input feature in the NN model.

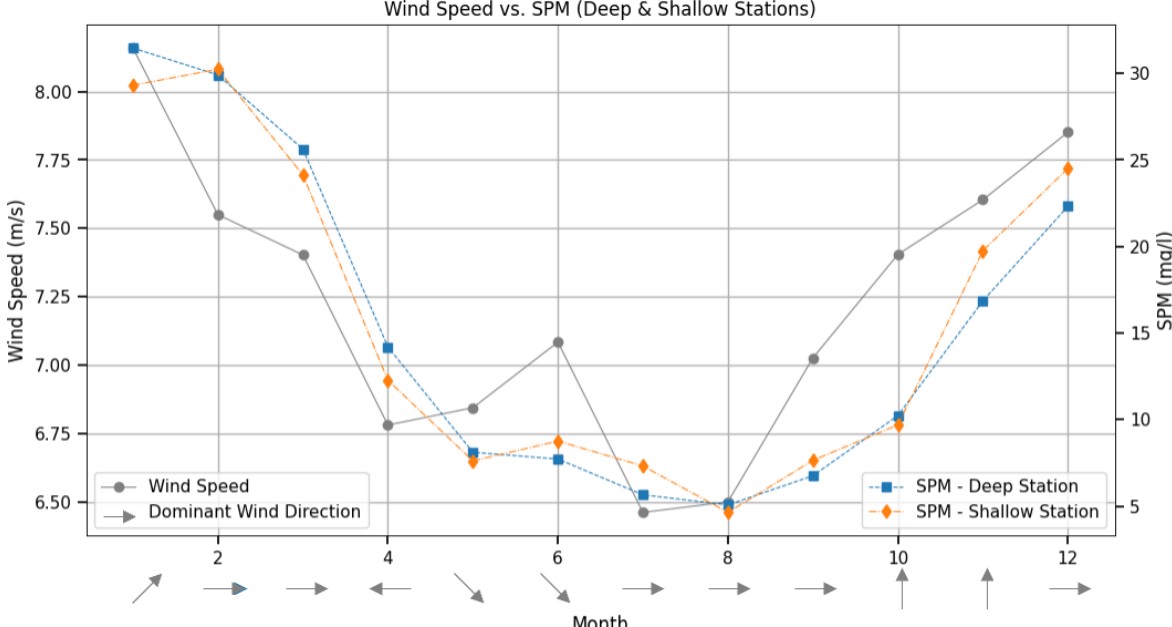

**Figure 5: Monthly averages of wind speed, SPM concentrations, and dominant wind direction (2000–2019). The figure illustrates**
**monthly mean wind speed (grey solid line, left axis) and SPM concentrations at the deep (blue squares) and shallow (orange**
**diamonds) stations (right axis). Grey arrows along the x-axis represent the dominant wind direction for each month, with arrow**
**orientation indicating the direction from which the wind originates, following standard meteorological convention.**
While correlations are further explored in Section 3.2, the seasonal alignment of wind intensity and
SPM concentrations suggests strong physical modulation.

## 3.2 Resuspension and Time Scales of Inner Basin Transport

Beyond seasonal variation, there are mechanisms responsible for the variability of SPM concentrations
on shorter temporal scales, from hours to days. This subsection examines how wind-induced
resuspension operates over different time frames and the role of tidal transport within the basin.

### 3.2.1 Role of wind forcing in short time SPM variations

To quantify the short-term response of SPM to wind forcing, we computed Pearson correlation
coefficients between SPM concentrations and wind speed averaged over different time intervals,
ranging from 1 hour to 240 hours. This approach, commonly referred to as "wind memory," allows us
to evaluate how the cumulative influence of past wind conditions affects SPM variability at different
depths and over varying time scales. Figure 6 illustrates how the strength of this relationship evolves as
wind speeds are averaged over progressively longer intervals, ranging from 1 hour to 240 hours. This
approach quantifies how SPM responds to the cumulative influence of past wind conditions over
varying time scales. The results show that correlation coefficients generally increase as wind memory
lengthens, reaching a peak around 12-18 hours at the shallow station and 120 hours at the deep station,
followed by a slight decline.
Monthly variations in the correlation patterns are provided in the Supplementary Material (Fig. S2),
showing that the correlation between wind speed and SPM is generally stronger in winter than in
summer, with more complex patterns during transitional months such as April and November.

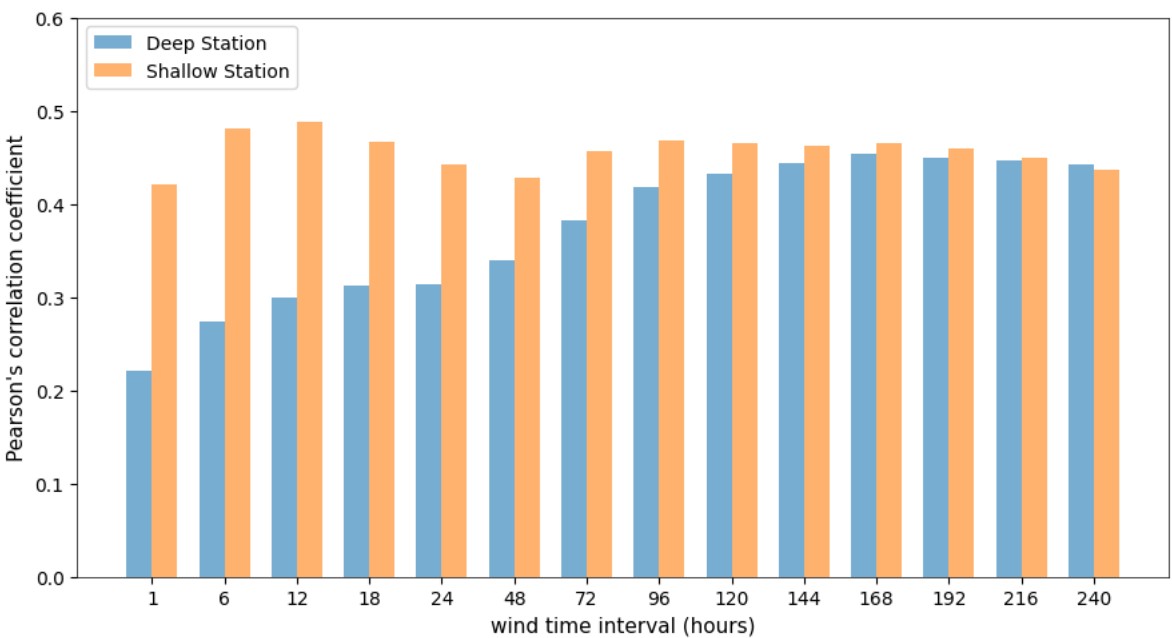

**Figure 6: Pearson correlation coefficients between SPM and wind speed averaged over different time intervals, ranging from 1**
**hour to 240 hours. The two stations (deep station as blue and shallow station as orange) are represented separately, showing how**
**wind memory influences SPM variability at different depths.**
The correlation between instantaneous wind speed and SPM concentrations is strongest during winter
and early spring (Fig. 7), particularly at the shallow station, where the most substantial value occurs in
January ($R^2 = 0.44$) and April ($R^2 = 0.50$). In contrast, the deep station shows much weaker correlations,
with $R^2$ values often near zero, reflecting a lagged effect on SPM concentrations.
During June–August, the relationship weakens significantly, with the lowest correlations observed in
August for both stations. The shallow station maintains minor correlation ($R^2 \approx 0.15$) into late summer
and early autumn.

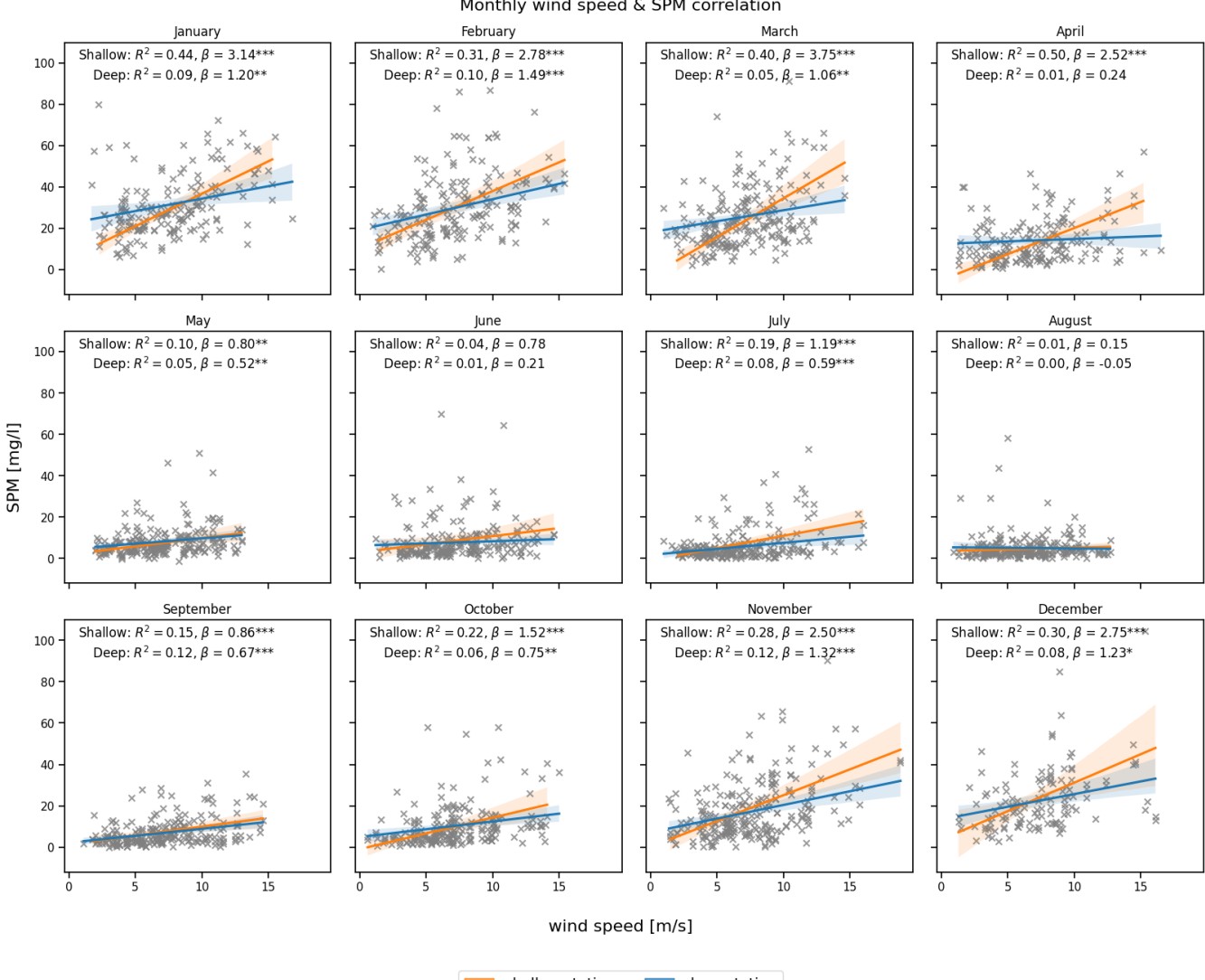

**Figure 7: Monthly correlation between wind speed at the time of sampling (x-axis) and SPM concentration (y-axis) at the deep (blue) and shallow (orange) stations. The coefficient of determination (R²), slope (β), and statistical significance (\*\*\*p < 0.001, \*\*p < 0.01, \*p < 0.05) are indicated in each panel.**

In contrast to the instantaneous impact of wind-speed, averaging wind speed over 120 hours (5 days) results in an improved correlation between wind and SPM at the deep station, as shown in Fig. 8, although the seasonal pattern remains similar. The stronger relationship is particularly evident in the winter and autumn months, especially at the deep station (e.g., $R^2 = 0.35$ in September, 0.24 in November and February). These findings indicate that the average wind-speed over several days have a stronger influence on SPM concentrations at the deep station than the wind at the time of sampling. The results highlight the role of wind forcing in modulating SPM concentration variability, particularly in shallow waters during winter and autumn. Naturally, the influence of wind is less pronounced in

deeper waters through direct resuspension mechanisms. However, it remains significant, with a time lag
possibly caused by the transport of resuspended material from shallower zones to the deep station.

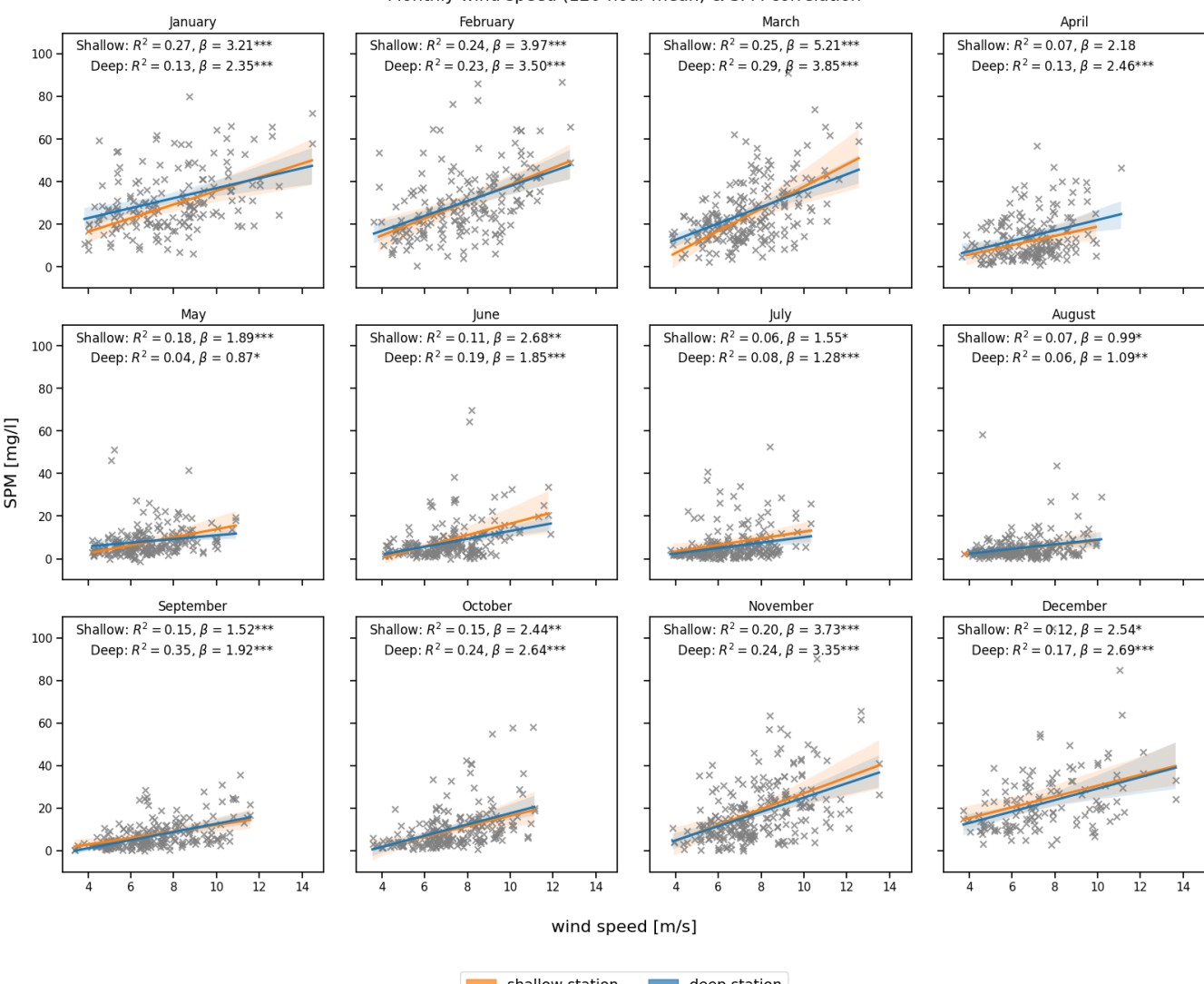

**Figure 8: Monthly correlation between wind speed averaged over a 120-hour (5-day) period (x-axis) and SPM concentration (y-**
**axis) at the deep (blue) and shallow (orange) stations. The coefficient of determination (R²), slope (β), and statistical significance**
**(\*\*\*p < 0.001, \*\*p < 0.01, \*p < 0.05) are indicated in each panel.**

**3.2.2 Role of tidal forcing in short time SPM variations**

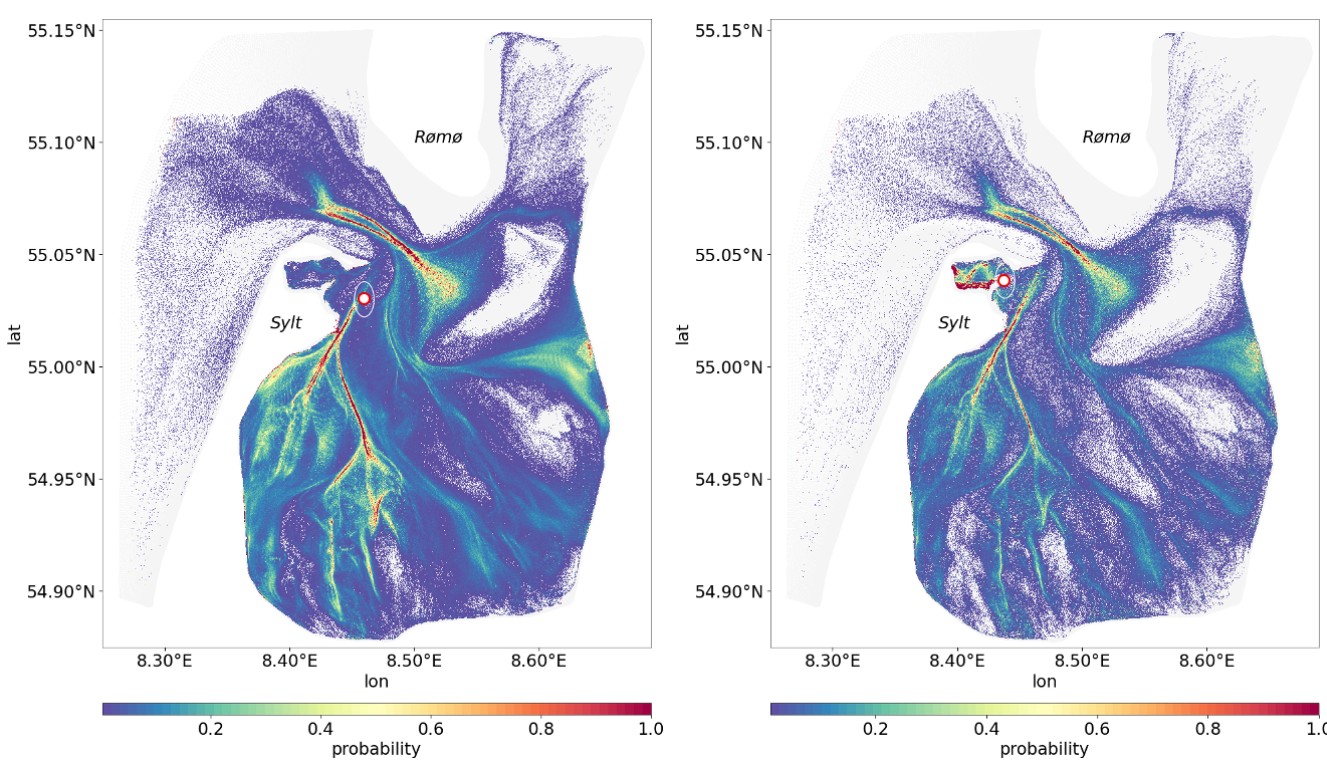

Figure 9: **Probability distribution of passive tracer pathways to illustrate how frequently different areas act as source regions or transport corridors for passive tracers. Higher probability values (highlighted in yellow and red) indicate locations that more frequently contribute to SPM reaching the deep station (left panel) or shallow station (right panel).**

Figure 9 shows the spatial probability distribution of passive tracer connectivity within the basin, derived from Lagrangian simulations forced solely by tides. This setup isolates the dominant physical driver of material transport in the Sylt-Rømø Bight, where tidal processes account for roughly 80% of velocity variability (Fofonova et al., 2019). While regular wind forcing (excluding storm events and uncommon multi-day episodes of strong winds from a single direction) enhances lateral dispersion, its net contribution to basin-wide transport over several days is relatively minor (Konyssova et al., 2025). Note, in this region, typical wind events last about 5–6 hours, defined as consecutive hours of wind from the same direction within an eight-sector classification (Rubinetti et al., 2023). Residual currents generated by non-linear tidal interactions establish consistent, directional transport pathways, which are effectively captured by the simulated tracer trajectories.

The resulting network of tidally-induced transport pathways reveals apparent differences between the two stations. The shallow station is predominantly supplied by tracers originating in the northern parts of the bight, particularly Königshafen and its surroundings, reflecting localized and relatively rapid connections. In contrast, the deep station is influenced by broader and more distributed transport pathways, integrating material from a wider area of the basin.

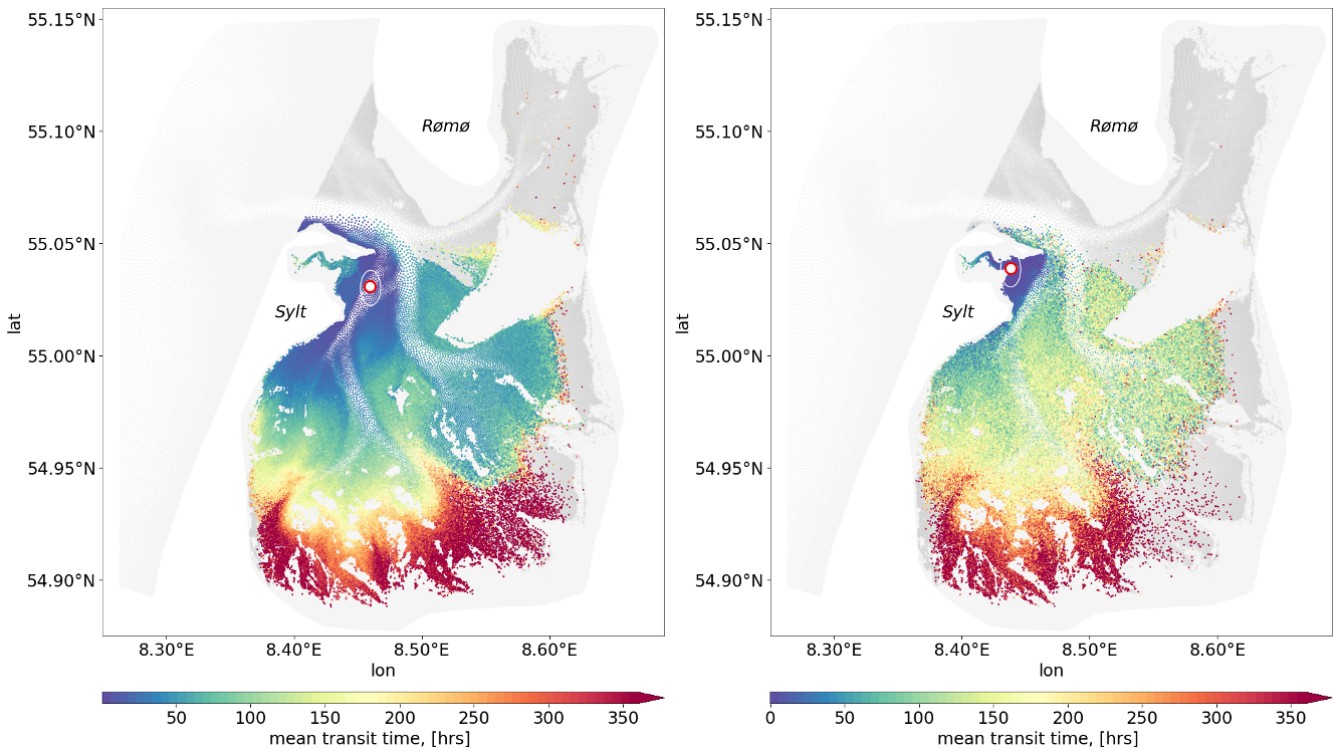

**Figure 10: The mean transit time of passive tracers to reach the sampling station. The left panel corresponds to the deep station,**
**while the right panel represents the shallow station. Grid elements that are consistently inundated with each flood phase, from**
**which the passive tracers are released, are shown in grey. Elements that successfully reach one of the two sampling stations are**
**colour-coded by their mean transit time (in hours), reflecting how long it takes for passive tracers to reach from a given location to**
**each station.**
For both stations, shorter transit times (depicted in blue and green, Fig. 10) are observed in areas closer
to the stations, whereas regions further away, particularly in the inner tidal flats, exhibit longer transit
times (yellow to red hues, Fig. 10). The transit time patterns reveal that SPM originating from intertidal
areas and tidal channels follows particular pathways before reaching the deeper and shallower
monitoring stations, with transport occurring on timescales of days to weeks. The difference between
the two panels suggests that the deep and shallow stations receive material from primarily distinct but
partially overlapping source regions.
Together, these figures illustrate the tidal connectivity of the two monitoring stations with their
surrounding basin, highlighting likely source regions and typical transport pathways. To quantify the
typical timescale for SPM transport, a probability-weighted transit time was computed, where
individual transit times were weighted according to their interpolated probability values. This approach
ensures that the estimated transit time reflects the most frequently occurring transport pathways rather
than rare, low-probability trajectories. The resulting probability-weighted median transit time was 133.3
hours for the deep station, which aligns with the ~5-day wind-memory interval identified in Section
3.2.2, and 44.4 hours for the shallow station, highlighting the pronounced contrast in transport
efficiency between the two locations and likely reflecting differences in sediment source regions.
Although not explicitly resolved in the simulations, this difference may also relate to grain-size
composition, as the deeper channels typically contain coarser material, whereas the intertidal flats have
finer, more cohesive sediments (Bartholomä et al., 2009). At the shallow station, which is situated
within the Königshafen area, SPM concentrations respond to wind forcing almost immediately through
local resuspension. The longer transit time therefore does not indicate a delayed response, but rather
reflects the additional supply of tracers arriving from a wider surrounding area, which extends the
median transit time beyond the short wind-driven response captured in the memory analysis. In contrast,
the deep station integrates SPM from more complex, multi-step transport routes over longer timescales.
These transit times provide a key reference for interpreting observed SPM fluctuations at each station
and may help distinguish between short-term and cumulative drivers of SPM variability.

## 3.3 Neural Network

### 3.3.1 Winter SPM prediction (Baseline Model)

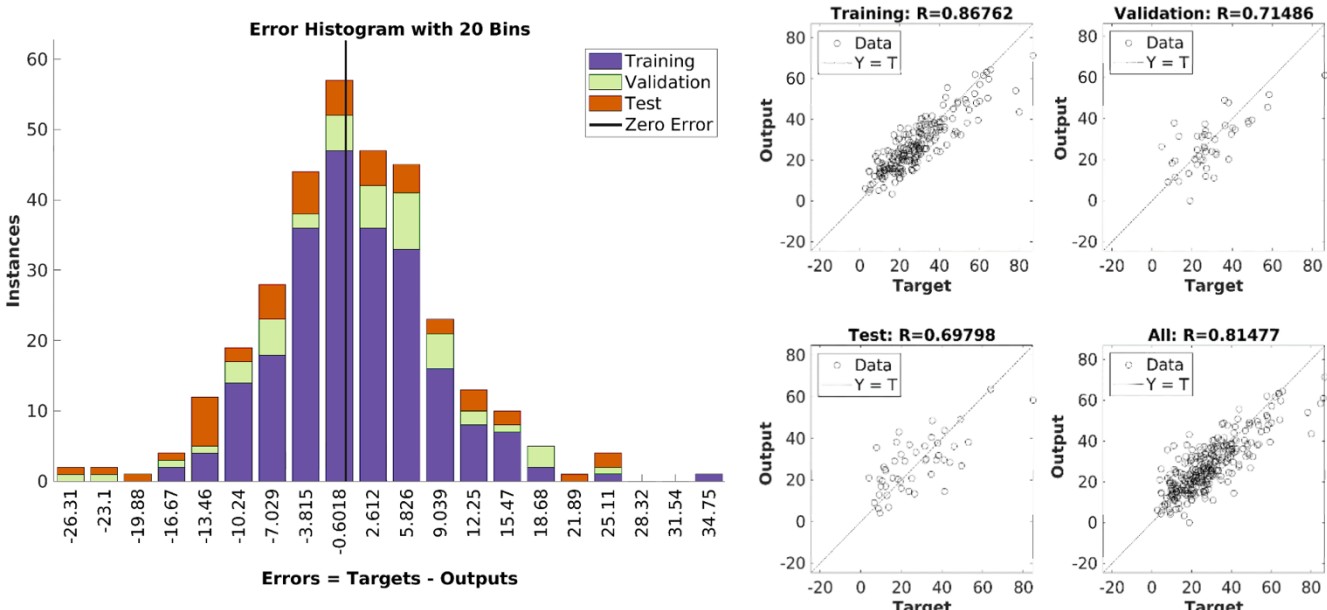

**Figure 11: Performance of the neural network (NN) trained on the winter dataset for the deep station. Left: Error histogram**
**showing the distribution of prediction errors (in mg/L) for training, validation, and test subsets. Right: Regression plots**
**comparing predicted vs. observed SPM values, with correlation coefficients (R) for each data subset. Due to the overall qualitative**
**similarity between the pictures for deep and shallow stations, only the deep station is presented here.**
To evaluate how well physical drivers alone can explain SPM concentration variability, we first trained
a baseline NN using only the winter dataset, when biological activity is minimal. This provides a
reference to assess the contribution of wind and tidal forcing under conditions with limited biological
influence.
The error for training, validation and testing has a normal distribution (Fig. 11, left). For the deep
station, the root mean square error (RMSE), a measure of average prediction error magnitude, is 9.4
mg/L for the testing set. Note that the observed mean and median SPM concentrations in winter are 27.6
and 25.9 mg/L respectively. As a result of the application of NN to the winter dataset, they are 27.2 and
25.5 mg/L respectively (Table 2). Regression analysis (Fig. 11, right) yields correlation coefficients
between observed and simulated SPMs equal to ~0.87, ~0.71, ~0.7 and ~0.81 for training, validation,
testing and all winter datasets, respectively.
The results show that, by using tidal and wind forcing along with a proxy for baroclinic conditions
(salinity), we can predict SPM concentrations during winter quite well without accounting for other
factors.
Notably, temperature, salinity, turbulent kinetic energy, and SPM concentrations themselves all
influence flocculation processes, even without considering biological activity. Temperature was
excluded from the feature set at this stage due to its strong seasonal cycle and its potential use as a
proxy for biological conditions. Specifically, in our current approach, we cannot separate the influence
of temperature on biological mechanisms from its physical effects (e.g. viscosity), which also impact
flocculation.

### 3.3.2 Applying of NN trained on winter data to other seasons

Next, we applied the NN trained under winter conditions to other seasons. This approach is justified by
the fact that the long-term dataset captures a representative range of wind conditions in both winter and
summer, including calm periods and strong wind events. Furthermore, salinity remains relatively stable
across seasons, supporting the applicability of the model. Using the trained winter model, we attempt to
predict SPM concentrations for spring, summer, and autumn. The goal of this step is to estimate the
influence of biological processes on SPM concentrations.
For both stations, the results of the NN application show significantly higher SPM concentrations
compared to observations (Table 2).
**Table 2. Mean, median values and correlation (R) of SPM concentrations, [mg/L], from observations VS predictions by NN**
**trained on winter dataset for the deep and shallow stations**

| seasons | | Deep Station | | Shallow Station | |
|---|---|---|---|---|---|
| | | observed | predicted | observed | predicted |
| winter | mean, [mg/L] | 27.6 | 27.2 | 28.5 | 24.7 |
| | median, [mg/L] | 25.9 | 25.5 | 24.4 | 26.5 |
| | | R = 0.81 | | R = 0.81 | |
| spring | mean, [mg/L] | 16.5 | 24.8 | 15.6 | 20.6 |
| | median, [mg/L] | 13.3 | 24.3 | 11.3 | 18.0 |
| | | R = 0.29 | | R = 0.5 | |
| summer | mean, [mg/L] | 6.15 | 17.7 | 6.8 | 18.5 |
| | median, [mg/L] | 4.22 | 16.9 | 3.9 | 16.1 |
| | | R = 0.04 | | R = 0.35 | |

| autumn | mean, [mg/L] | 11.6 | 23.4 | 12.8 | 24.0 |
| | median, [mg/L] | 8.5 | 22.7 | 7.5 | 21.0 |
| | | R = 0.44 | | R = 0.54 | |

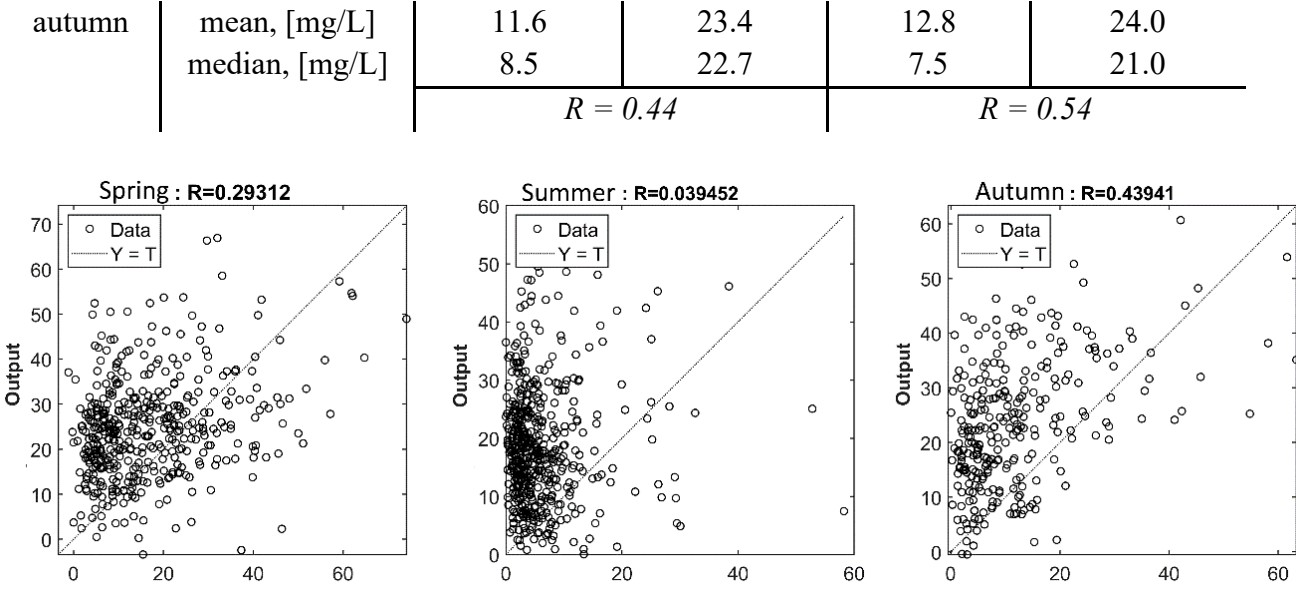

**Figure 12: Regression analysis for NN trained on winter data and applied to spring, summer and autumn data vs observations. Results shown for the deep station; shallow station results are qualitatively similar and summarized in Table 2.**

This discrepancy suggests that while calmer wind conditions alone can explain approximately 40% of the SPM reduction in summer compared to winter, they do not account for the full ~80% decrease observed in the data (see Table 2). The overestimation of SPM by the NN is clearly visible in the regression plots (Fig. 12), where most of the predicted values are positioned above the Y = T line, particularly during summer and autumn. This does not necessarily imply that biological activity is weaker in spring, but rather that its impact is less well captured by the abiotic-only model, as biological processes contribute more strongly to variability than in autumn. The higher correlation between simulated and observed SPM in autumn likely reflects physical conditions that are more similar to winter, when the model was trained, including generally stronger winds (Section 3.2.2). In spring, enhanced biological activity and aggregation introduce variability that the model cannot account for, reducing correlation. During summer, the regression coefficient dropping below 0.05 indicates that processes not represented in the abiotic model become dominant (Fig. 12). The subsequent analysis of all seasons presented in Section 3.3.3 points to biologically mediated mechanisms, such as flocculation and sediment stabilization, as the main contributors to this variability.

### 3.3.3 NN trained at the data from all seasons

Previously we showed that there are factors, in particular biological processes, which significantly influence the SPM dynamics. To confirm this, we trained NN on all the dataset (not limited to winter) using the same input features representing only abiotic conditions, which we used in 3.3.1. As expected, the overall RMSE increases to approximately 11 mg/L, due to the omission of important features that influence SPM dynamics during the warm season (Supplementary Fig. S5). When the model is trained on data from all seasons, and applied specifically to winter conditions, it

underestimates SPM concentrations. For the deep station, the predicted mean and median are 19.17
mg/L and 17.2 mg/L, respectively, compared to observed values of 27.6 mg/L and 25.9 mg/L. At the
shallow station, the model yields mean and median predictions of 20 mg/L and 17 mg/L, whereas the
corresponding observed values are 28.45 mg/L and 24.35 mg/L. The regression coefficients also reflect
this underperformance, with values of approximately $R \approx 0.66$ (deep) and 0.72 (shallow). The seasonal
regression analysis further highlights these biases (Supplementary Fig. S7, upper row). A complete
numerical breakdown of observed and predicted seasonal values is provided in Table S3.
**3.3.4 Neural network trained on all-season data including biological proxy features**
Including temperature and sunshine duration as proxies for biological activity improved overall model
skill, reducing the test RMSE to ~10 mg/L. The regression analysis shows the correlation coefficient
between observed and simulated SPM concentrations reached ~0.88, ~0.76, ~0.72 and ~0.84 for
training, validation, testing and all datasets respectively (Supplementary Fig. S6).
Seasonal regression plots also show marked improvements (Supplementary Fig. S7, lower row). For
winter, R increased from 0.66 to 0.73 at the deep station and from 0.72 to 0.82 at the shallow station.
The largest gains occurred in spring and summer, consistent with more substantial biological influence
during these seasons. Table S3 provides the detailed seasonal comparison of observed and predicted
means, medians, and correlations.
**4 Discussion**
This study set out to investigate the combined influence of wind forcing, tidal transport, and biological
processes on SPM concentration variability in the Sylt-Rømø Bight. Using 20-year period monitoring
data and applying statistical tools, neural network models, and Lagrangian simulations, we analyze the
relative roles of these processes across seasonal and short-term timescales.
The variability observed in SPM concentrations reflects the inherent complexity of the system, where
several physical and biological mechanisms interact across different spatial and temporal scales. Wind-
driven resuspension, tidal advection, and biologically mediated effects each contribute to the observed
dynamics, with their relative importance shifting seasonally. This complexity highlights that the
variability in SPM concentrations cannot be attributed to a single driver but rather to the combined
response of the system to multiple, overlapping processes.
In this chapter, we will further discuss topics that have not yet been fully addressed: the peculiarities of
the Chl-a and SPM concentration relationship, biological processes that directly or indirectly influence
SPM dynamics, further details on wind and tide control including SSH and SPM concentration
relationship, details of NN related findings and study limitations.
**4.1 Biological Interaction**
The strong positive correlation between Chl-a and SPM concentrations in winter suggests that under
low biological activity, such as limited phytoplankton growth, behaviour of both variables is driven
mainly by physical processes such as wind-driven resuspension. This is consistent with previous

observations in the Wadden Sea and German Bight (van Beusekom et al., 1999; van Beusekom and de Jonge, 2002). Together with SPM, the strong winds resuspend microphytobenthos attached to sediment particles (de Jonge and van Beuseukom, 1995), consistent with observations by Hommersom et al. (2009), who found strong correlations between SPM and Chl-a in the Wadden Sea driven by resuspension. Such patterns support the interpretation that, under winter conditions with low biological production, a large fraction of Chl-a is contained within resuspended flocs, leading to the high correlations observed in our study at both stations. In contrast, the correlation between Chl-a and SPM concentrations begin to decline in spring, despite persistent wind forcing, and reaches a minimum in summer. This seasonal decoupling suggests that biological aggregation processes become increasingly dominant with the beginning of and following phytoplankton bloom, promoting the formation and subsequent rapid settling of flocs (Schartau et al., 2019; Maerz et al., 2016; Lunau et al., 2006). For example, the production of TEPs, secreted by phytoplankton and bacteria under certain physiological or nutrient conditions, strongly enhance particle stickiness and promote the aggregation of both organic and mineral particles (Passow, 2002). While our statistical analysis (however, not NN related efforts) uses Chl-a as a proxy for biological activity, it is worth noting that TEP production responds to additional drivers such as nutrient status and species composition, which can modulate flocculation potential even at similar Chl-a levels. In addition to the biological feedback, the increase in water temperature during summer also accelerates particle sinking rates by reducing water viscosity, further promoting sediment deposition (Maerz and Wirtz, 2009). This supports the view that the Chl-a–SPM relationship reflects an interplay of multiple biological and physical controls rather than a single, direct pathway.

While both spring and autumn are characterized by peaks in Chl-a concentration, their influence on SPM dynamics appears to differ. In spring, calmer conditions favor particle aggregation and settling, often leading to reduced SPM levels. In autumn, however, the secondary bloom coincides with a seasonal increase in wind speed (Fig. 5), resulting in elevated SPM concentrations despite high biological activity. This suggests that enhanced organic matter availability in autumn may promote flocculation, but stronger resuspension prevents effective settling. These seasonal contrasts highlight that biological influence on SPM is strongly modulated by concurrent physical forcing.

In parallel, benthic microbial processes such as biofilm formation by microphytobenthos enhance sediment stability and reduce its susceptibility to resuspension, thereby modulating the amount of SPM remaining in the water column (Andersen, 2000; Stal, 2010). Beyond microbial stabilization, benthic fauna also influences SPM dynamics through benthic-pelagic coupling. Filter-feeding benthos, such as bivalves, consume suspended particulate organic carbon (POC), effectively removing material from the water column and altering vertical fluxes of organic matter. In the Sylt-Rømø Bight, food web modelling by Baird et al. (2004) estimated that benthic consumers remove approximately 56.7 mgC m$^{-2}$ d$^{-1}$ of suspended POC – potentially a substantial fraction of total SPM, depending on the proportion of organic material within it. Benthic consumption likely constitutes a significant sink for fine, organic-rich particles, particularly in areas with dense benthic communities, as filter feeders remove SPM primarily through ingestion and the production of faecal pellets, which promote deposition of organic-rich material to seabed. Although the proportion of POC within SPM is known to depend on total SPM concentration (Schartau et al., 2019), robust SPM-to-POC conversion factors remain uncertain due to the unknown share of freshly formed organic matter. Although the effects of native species on bentho-

pelagic coupling have been relatively well studied in the region (Baird et al., 2007), the role of
introduced species remains less clear. For example, the introduction and spread of alien species such as
the American razor clam (*Ensis leei*) may further amplify benthic filtration effects. In the Dutch
Wadden Sea, they have shown to significantly alter trophic carbon flows, increasing carbon
consumption by secondary producers and redirecting energy flows away from higher trophic levels
(Jung et al., 2020). Although the abundance of *E. leei* in the Sylt-Rømø Bight has not yet been
quantified, its potential influence on observed SPM concentrations provides a basis for further
investigation as part of long-term ecological changes in the region.

## 4.2 Wind & Tide Control

The results suggest that the intensity of winds, rather than their dominant directions, exerts a stronger
influence on the seasonal cycle of SPM. The seasonal patterns of wind speed, with maximum values in
winter and minima in summer, aligns closely with the observed SPM concentrations, suggesting a
physically mediated control during high-energy periods. During summer, the weaker relationship
between wind speed and SPM concentrations points to the increasing importance of biological
contribution. These may include enhanced flocculation and settling of particles, increased filtration by
benthic organisms, reduced sediment availability due to biological stabilization, and overall calmer
wind conditions. Although tidal forcing itself does not vary seasonally, its influence on SPM
concentrations also likely changes throughout the year because the sediment properties and biological
activity that determine sediment availability and erodibility vary over time.
Notably, correlations between wind speed and SPM are stronger at the shallow station compared to the
deep station, as the shallow environments are more susceptible to wind-induced turbulence, including
wave action. It should be noted that, since the study addresses observations of SPM concentrations,
wind-induced resuspension is considered to include the effects of wave action.
In addition, the area around the shallow station is characterized by relatively small depths. Therefore, at
the shallow station, the short wind-memory intervals (12–18 hours) yield the strongest correlations.
Lagrangian simulations support this interpretation by illustrating that the shallow station, located within
the intertidal zone itself, receives SPM directly from resuspension at the site. While these simulations
do not represent full sediment dynamics, they effectively illustrate dominant source areas and transport
timescales. This relatively short response time aligns with earlier observations in tidal flat systems,
where wind effects on SPM are immediate in intertidal zones but take longer to become apparent in
deeper areas (de Jonge and van Beusekom, 1995).
In contrast, the situation at the deep station is more complex. In addition to the larger distance between
the observation point and the sea bottom, sediment characteristics in the channels are predominantly
sandy, which typically requires stronger wind energy to be resuspended as opposed to the mud and finer
material found in intertidal areas (Bartholomä et al., 2009). The delayed response seen in the wind
memory analysis (~120 hours) and the tidally modulated transport pathways from our Lagrangian
simulations (median transit time ~133 hours) likely reflects the combined influence of episodic wind-
driven resuspension in shallow areas and subsequent advective transport to the deeper station
(Friedrichs and Perry, 2001; de Jonge and van Beusekom, 1995). Similar multi-day lags in SPM
response have been observed in other estuarine systems where tidally dominated transport pathways act
on resuspended sediments originating from more energetic, nearshore zones (see also Winterwerp,
2001). Together, these results highlight a clear contrast between the two stations: rapid, localized
sediment response in the shallow embayment versus slower, more integrated transport processes at the
relatively deep channel.
The Lagrangian tracer simulations provide additional spatial context, revealing weak connectivity
between the north-eastern and southern sections of the Sylt-Rømø Bight under regular tidal conditions.
Passive tracers released in the north-eastern region rarely reach either of the sampling stations. This
indicates that the stations, while valuable for long-term monitoring, may not fully capture the spatial
variability of SPM dynamics across the entire basin, especially on shorter time scales.
Beyond local wind and tidal resuspension and transport within the Sylt-Rømø Bight, tidal phases also
influence SPM concentrations. Naturally, higher SPM concentrations are associated with lower water
levels under otherwise equal conditions, as the measurements are always taken at the surface (Fig. S3).
Due to the large role of non-linear processes in the area there is also an asymmetry between flood and
ebb in terms of duration as well as mean and maximum velocities.

### 4.3 Neural Network findings

The NN experiments provide a line of evidence supporting the seasonal shift in dominant SPM drivers
independent of the statistical analysis. When trained only on winter data, where biological activity is
minimal, the NN successfully captured winter SPM concentrations using solely physical drivers such as
wind conditions within different time intervals, salinity, and tidal elevation (through SSH and SSH
gradient). At both stations, this winter model performed well ($R \approx 0.81$), confirming that physical
processes alone can account for most of the observed variability during this season. This reinforces the
view that winter dynamics are dominated by abiotic drivers.
However, when the same winter-trained model was applied to other seasons, it consistently
overestimated SPM concentrations, especially in summer when biological activity is high and wind
speed is generally weaker. For example, at the deep station, the observed mean SPM concentration
decreased by ~78%, whereas the model reproduced only a ~36% reduction, meaning that less than half
of the observed seasonal decline could be explained by abiotic factors. Similarly, at the shallow station,
observed SPM declined by 80%, while the model captured only a ~42% reduction, accounting for just
over half of the change. These mismatches underscore the increasing importance of biological drivers
during the summer months. Interestingly, although both stations showed similar magnitudes of
overestimation, the shallow station retained higher predictive skill ($R \approx 0.35$) than the deep station ($R \approx$
$0.04$) in summer. This difference likely reflects the stronger and more immediate influence of wind
forcing at the shallow site, where even in summer, intermittent wind events can rapidly mobilize
sediments.
The addition of biologically relevant features such as temperature and sunshine duration into the full-
year NN model, in Section 3.3.4, significantly improved its performance, especially at the shallow
station. While we do not explicitly resolve individual biological pathways, these proxies likely capture
their cumulative effects as seen from the improved model fit indicating their aggregate influence is both
detectable and substantial. These processes may include phytoplankton-driven aggregation and
flocculation, microbial stabilization, and bentho-pelagic interactions such as benthic filtration. Our

interpretation remains exploratory rather than conclusive, as targeted biological data would be required to quantify these mechanisms in detail.

**4.4 Study Limitations**

Although, in this study, we discuss the main driving mechanisms of SPM concentrations in the Sylt-Rømø Bight, we also acknowledge that the system is far more complex and has more factors influencing the fluctuations of SPM concentrations than we investigate here. Moreover, the SPM itself is a parameter that is composed of various components and sizes in nature, and such a level of detail is not readily available in the dataset. Incorporating measurements of sediment composition (organic versus inorganic fractions) and their sizes would add more certainty into the roles of considered mechanisms modifying SPM concentrations.
Our Lagrangian transport simulations use massless passive tracers, which do not account for flocculation, deposition, or resuspension processes. Incorporating such processes would require a different modeling framework and additional data that are not currently available, including detailed habitat maps and reliable estimates of SPM fluxes from the open boundary. Nonetheless, despite these simplifications, the model effectively captures key patterns of spatial connectivity pathways and timescales driven by tidal dynamics.
Another limitation is that changes in the SPM input from the open boundary cannot be quantified. Net coastward transport from the North Sea likely contributes to seasonal variability, for example, through enhanced import or retention in spring and summer (Burchard et al., 2008; Postma, 1981), but we lack data to resolve this numerically.
In addition, more precise SPM predictions would benefit from neural network architectures with "memory", such as Long Short-Term Memory (LSTM) models, which can account for the delayed effects of biotic and abiotic conditions over several months. A North Sea-wide approach would be required to capture these long-term dynamics and boundary-driven influences. Within such a framework, it would also be appropriate to include additional predictors such as nutrient concentrations and benthic processes to represent better the complex interactions driving SPM variability.
While this study does not claim to offer a comprehensive representation of all the processes in play, we believe that our work presents new insights to understand better the baseline mechanisms of SPM concentration variability within the basin and across multiple timescales.

**5 Conclusions**

This study investigated the primary drivers of SPM variability in the Sylt-Rømø Bight, a semi enclosed basin with well-mixed conditions and hydrodynamics dominantly shaped by tides. We combined long-term monitoring data (SPM, Chl-a, wind conditions, and light availability) with SSH reconstructed from a validated hydrodynamic model, and applied statistical analyses and neural network modelling. In addition, Lagrangian transport simulations were used to assess tidal connectivity and transport timescales. Together, these approaches revealed new insights on SPM concentration variability across multiple timescales and relative influence of the main driving mechanisms.

Our findings confirm that wind speed is the primary driver of short-term SPM variability, particularly at the shallow station, where SPM concentrations respond almost immediately to wind forcing. This rapid response, especially during winter and autumn, highlights the importance of wind-driven resuspension in shallow waters. In contrast, the deep station exhibits a more delayed response to wind forcing, with peak correlations occurring at longer wind memory intervals (~5 days). This lag reflects the fact that, at greater depths, direct resuspension due to immediate wind forcing plays a reduced role, while the transport of material from neighboring shallower areas becomes increasingly important.

Tidal dynamics primarily regulate the advection processes within the basin, redistributing fine, easily resuspendable material from shallow to the deeper areas. Lagrangian simulations illustrate that SPM at the shallow station originates locally, predominantly from within or around the Königshafen embayment. Meanwhile, at the deep station, SPM is likely supplied from the intertidal and shallow areas by the tidally driven redistribution over a longer timescale (~133 hours), consistent with the observed wind-memory lag (~120 hours). These results highlight the fundamental distinction between localized, wind-driven resuspension and slower, broader-scale, tide-driven transport, both of which shape SPM variability but at different spatial and temporal scales.

On seasonal timescales, however, biological processes exert an increasingly strong control. The onset of the spring phytoplankton bloom coincides with a decline in SPM concentrations, likely due to the enhanced production of TEPs and other extracellular polymers that promote biological aggregation and flocculation, leading to enhanced particle settling. This strengthening of biological effects reduces the efficiency of physical resuspension even under comparable wind conditions, which results in the pronounced decline in SPM concentrations toward summer. The inverse seasonal patterns of Chl-a and SPM support this interpretation, aligning with previous studies that describe the role of phytoplankton in promoting flocculation and sedimentation in coastal systems. NN experiments suggest that calmer wind conditions alone can explain approximately ~40% of the observed summer SPM reduction compared to winter concentrations. Still, they do not account for up to ~80% decrease seen in the data. This substantial reduction is likely influenced by a variety of biologically related mechanisms, ranging from the microbial activity, production of EPS to zooplankton grazing. Further studies are needed to quantify the relative contributions of these individual mechanisms.

Overall, this study provides a comprehensive and quantitative assessment of how wind, tides, and biological activity interact to control SPM variability in a shallow, tidally dominated coastal system.

**Conflict of Interest**

The authors declare that they have no conflict of interest.

**Author Contributions**

GK performed the data analysis and wrote the initial draft of the manuscript. GK, VS, and AA carried out the numerical simulations. JvB, GK, and VS conceptualized the study. SH, SR, IK, and KHW contributed to the discussion of methods and interpretation of the results. All authors contributed to manuscript review and editing.

**Funding**

This study has been funded by the German Federal Ministry of Education and Research (BMBF) in the frame of the joint research projects MGF-Nordsee (FKZ 03F0847A), CREATE (03F0910B) and Coastal Futures (FKZ 03F0911J) part of the research mission "Protection and Sustainable use of Marine Areas", within the German Marine Research Alliance (DAM).

**Data Availability**

The source code of the FESOM-C model is publicly available via Zenodo: https://doi.org/10.5281/zenodo.2085177 (Androsov et al., 2018). The Sylt Roads observational dataset is accessible through the PANGAEA data portal: https://doi.org/10.1594/PANGAEA.873549, https://doi.org/10.1594/PANGAEA.873547, https://doi.org/10.1594/PANGAEA.918018, https://doi.org/10.1594/PANGAEA.918032, https://doi.org/10.1594/PANGAEA.918027, https://doi.org/10.1594/PANGAEA.918023, https://doi.org/10.1594/PANGAEA.918033, https://doi.org/10.1594/PANGAEA.918028, https://doi.org/10.1594/PANGAEA.918024, https://doi.org/10.1594/PANGAEA.918034, https://doi.org/10.1594/PANGAEA.918029, https://doi.org/10.1594/PANGAEA.918025, https://doi.org/10.1594/PANGAEA.918035, https://doi.org/10.1594/PANGAEA.918030, https://doi.org/10.1594/PANGAEA.918026, https://doi.org/10.1594/PANGAEA.918036, and https://doi.org/10.1594/PANGAEA.918031. Meteorological data, including hourly wind characteristics (station 3032, List auf Sylt; dataset ID: *urn:x wmo:md:de.dwd.cdc::obsgermany-climate-hourly-wind*) and daily sunshine duration (*urn:wmo:md:de-dwd-cdc:obsgermany-climate-daily-kl*), are available from the Climate Data Center (CDC) of the Deutscher Wetterdienst (DWD): https://opendata.dwd.de/climate_environment/ (last access: 28 February 2025). The Lagrangian model output and Neural Network results are available from the corresponding author upon reasonable request.

**Acknowledgements**

We thank the teams involved in the Sylt Roads long-term ecological monitoring program for providing essential in-situ data. We also acknowledge the Deutscher Wetterdienst (DWD) Climate Data Center for access to freely available meteorological datasets. This study was conducted as part of the research mission "Protection and Sustainable Use of Marine Areas" of the German Marine Research Alliance (DAM) and was financially supported by the German Federal Ministry of Education and Research (BMBF).

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
