# Peer review of "Wind and Phytoplankton Dynamics Drive Seasonal and Short-Term"

_EGUsphere, 2025_

## Referee Comment (RC2)

**Review** of the manuscript *Wind and Phytoplankton Dynamics Driev Seasonal and Short-Term Variability of Suspended Matter in a Tidal Basin* , by Konyssova et al, submitted to **Biogeosciences** (egusphere-2025-2135).

**Manuscript overview**

The manuscript combines evidence from observational correlations, particle tracking and a neural network model to derive likely drivers for observed SPM dynamics in the Sylt-Rømø tidal inlet located in the German/Danish Wadden Sea. The correlations from 2 long term stations in the inlet are primarily used to illustrate a decrease in correlation during certain seasons, showing stronger correlations during winter. The particle tracking is applied to identify regions within the inlet that supply sediments to the 2 observational stations by tidal processes, demonstrating a local supply of sediment to the shallow station and a (Sylt) basin-wide supply of sediment to the deep station. The neural network model is applied using varying predictors and seasonal training sets, displaying decreasing winter correlation values when all seasons are used for training and increasing winter correlation when biological proxies are included, all relative to the winter trained abiotic NN model application to winter observations. The authors conclude that wind time lag is important for the sediment supply to the deep station while wind is important for the shallow station, that physical drivers determine the main winter sediment dynamics and that in summer additional biological controls account for roughly half of the 80% observed reduction in SPM levels, attributing the other half to wind reduction.

**Review overviews**

The manuscript is well written (though a bit repetitive in places) and has many supporting graphs. The study is essentially very local but has useful conclusions for similar systems, at least throughout the Wadden Sea. The main conclusions are not new but quantify the separate contributions from the separate drivers for this particular inlet during winter and summer. The approach is novel, combining observations, particle tacking and a neural network (NN) model to identify changes in (usually) already weak corelations. Personally I found the particle tracking study the most interesting part, and a good way to identify source regions without doing a full morphodynamic study (and truly considering SPM dynamics). The manuscript lacks some information in places (like the predictors included in the NN), seems to contain some errors in listing results, does not always include enough evidence for statements, does not specifically reference the supplementary materials and relies in many parts on weak and even weaker correlations. The latter I consider a suboptimal bid for information, but as sediment resuspension is a notoriously difficult process to predict given the large range of spatial and temporal scales involved I do find it understandable and, to be fair, a brave choice. Overall the authors make a relatively convincing case for the identified drivers and their seasonal impact, though they should be cautious with their conclusions when considering observational correlations and biological proxies. They are however much more cautious with these results in their discussion and conclusions. More detailed comments are listed below.

**Recommendation**

Major revision, though new simulations are not necessary.

**Detailed Comments**

1.  Line 56: I get the point but the wording is awkward, suggesting there is a sediment balance to begin with.

2. Section 2.1: in the area description I miss mention of possible anthropogenic activity in the region that could influence observed SPM levels. Is there local ship traffic (commercial, recreational), ferry service, dredging or trawling activity?

3. Line 147: I am a little bit surprised that M6 is not included here, but have not checked the listed references for its importance.

4. Line 153: please be table or figure specific when referring to the supplementary materials. This applies to the whole manuscript.

5. Line 161: needs rephrasing, e.g. "calculated … were released" or 'We released …'

6. Line 233: here winter is mentioned for the first time outside of the abstract and introduction, so I would expect a definition of the seasons used here. Note that different research fields can have different delineations of the year into seasons. The Wadden Sea Quality Status report of 2017 even used January-March for winter (van Beusekom, 2017), and astrological definitions do not follow whole months. So please specify which delineation was used in this work.

7. Line 239: I'm not sure what is meant here with "current elevation", but I assume the authors mean SSH? And a table overview of the predictors of the NN model is necessary, likely in the supplementary material. Counting the listed predictors here I find 5 directly mentioned predictors or 19 when counting the individual lag components separately. But not the 21 mentioned in 3.3.3 for the non-biotic version of the NN. Please list the basic and added (biological proxy) NN model predictors clearly.

8. Figs 4, 6, 7: these could go into the supplementary materials with simple a table in the main text listing the correlation metrics in one table for all figures. This would facilitate comparison of the individual month correlations over the different observation drivers with observed SPM.

9. Line 250: I miss references here to more work regarding EPS, its production by phytoplankton and its role in flocculation.

10. Line 274: the data also shows an autumn bloom which seems to have a different impact on SPM dynamics compared to the spring bloom mentioned here. Some explanatory text on this phenomenon (why is there an autumn bloom in this shallow inlet and why does SPM increase at the same time? ) would be useful here.

11. Line 291: the use of "*winter (December-January) … autumn (August-October)*" suggest a seasonal definition that has not been specified before and which suggest unequal length of the seasonal delineation. And which is at odds with other parts of the manuscript (e.g. line 299 where winter is suddenly December-February, while in line 312 late autumn consists of November and December but in line 328 it is listed as October-November).

12. Line 302: the listed do not match those in fig. 4 where December has 0.84 at the shallow station and January has 0.75 at the shallow station.

13. Line 306: all $R^2$ are < 1 as $R^2$=1 would be a perfect fit. So not sure why this is included here.

14. Line 309: please include a list of predictors used in each separate NN model application.

15. Line 339: $R^2$=0.44 is by no means a strong correlation and it is not even the highest correlation found as April lists $R^2$=0.50 for the shallow stations correlation between wind speed and SPM levels. And it should be mentioned clearly in the text that a correlation between 2 timeseries is by no means a causal relationship, as demonstrated by the strong correlation between winter Chl-a and SPM levels. Therefore I find line 341 rather contentious: it was simply tested here if there was a correlation between 2 timeseries, and for January only 44% of the SPM variability was explained by the instantaneous wind speed and in April 50%. Hardly a dominant control in winter I think. In my opinion the authors should be more careful in their statements regarding the observational time series correlations.

16. Line 353: "*The stronger relationship*" is more accurately a less weak relationship.

17. Line 357: I would say that wind plays a role in controlling SPM variability on different temporal scales, but a crucial role suggests higher correlation factors than have been found here. Besides, the authors only consider tides in the particle tracking study because they are the dominant resuspension mechanism (line 428), and do not include wind waves here.

18. Line 373: I'm sorry but I don't see this in Figure 8 (rather the opposite) and any kind of metrics to support this statement are missing. Please include more supporting evidence here.

19. Line 515: a reference to Table 2 is missing here as a source for the 80% reduction mentioned.

20. Lines 518-520: there is no reference here to figure S1 and the supplement does not contain any metrics for a direct comparison of seasonal wind speed. Indeed, autumn winds look a lot stronger in S1 than the spring or summer winds. The figure quality of S1 is too low as well. But I see no evidence for the statement in these lines.

21. Lines 520-522: or it speaks to the stronger winds in autumn?

22. Line 523: tidal dynamics was already mentioned as the dominant driver of SPM conditions, now biological processes are mentioned here as being dominant in summer. I would suggest that biological processes become the primary driver of SPM variability in summer, not of SPM dynamics.

23. Section 3.3.3: it would be good to include seasonal plots for the 2 additional NN model applications (all seasons physics and all seasons plus biological proxies) in the supplementary materials. That is the equivalent of figure 14.

24. Section 3.3.4: the title is not semantically correct.

25. Line 542: please reference figure S2 directly.

26. Lines 553-554: Amen to that. I think the statement here is more accurate than other statements throughout the manuscript mentioning crucial or dominant drivers. And the authors should not sell themselves short: this is not new in itself but they have helped quantify the parts played by different drivers for this inlet and for these data years.

27. Section 4.1: I miss a more detailed discussion about the biologically induced flocculation process here, as for instance TEP (transparent exopolymer particles) production occurs under certain circumstances and therefore cannot be predicted by Chl-a levels.

28. Line 571: I miss a reference here to Hommersom et al (2009) who report a correlation between SPM and Chla due to resuspension in their work. A discussion of this work and the difference with the present study should be included.

29. Line 590: "*constrained*", do you mean unknown?

**References**

van Beusekom, J.E.E., Bot, P., Carstensen, J., Grage, A., Kolbe, K., Lenhart, H.-J., Patsch, J., Petenati, T., Rick, J. (2017), Eutrophication, in the Wadden Sea Quality status report, version 1.01 (https://qsr.waddensea-worldheritage.org/reports/eutrophication)

Hommersom, A., Peters, S., Wernand, M. R., & de Boer, J. (2009). Spatial and temporal variability in bio-optical properties of the Wadden Sea. *Estuarine, Coastal and Shelf Science*, *83*(3), 360-370.

---

## Author Comment (AC1)

**Response to Reviewer #3**

**Manuscript title:** *"Wind and Phytoplankton Dynamics Drive Seasonal and Short-Term Variability of Suspended Matter in a Tidal Basin"*

We are grateful to Reviewer #3 for the careful and constructive review. The comments raised important points that helped us clarify the design of our experiments and the scope of our conclusions. Below, we address each comment point by point and outline the corresponding revisions implemented in the manuscript.

Reviewer comments are shown in **bold**, and our responses are given in regular font below each comment.

..............................................................................................................................................

Review of "Wind and Phytoplankton Dynamics Drive Seasonal and Short-Term Variability of Suspended Matter in a Tidal Basin" by Konyssova et al. (egusphere-2025-2135)

**This is a long paper, trying to examine the seasonal and short-term variation of suspended matter in a tidal basin. Both numerical model and neural network are used in this study. However, it is very confusing to this Reviewer why the experiments are designed in this way. Do you really need both models?**

The research is based on observational data collected over the past 20 years at two fixed locations. During this period, a wide range of possible forcing scenarios occurred. The typical duration of a wind event, which is defined as a continuous number of hours when wind blows from one direction, is about 5 hours, divided into eight sectors and utilizing hourly data (Rubinetti et al., 2023). To capture all these scenarios and the non-linear interplay between tidal and wind forcing that determines SPM concentrations, we applied a relatively deep Neural Network (NN). Using purely numerical simulations would not allow us to represent such a broad range of scenarios. Moreover, it was not feasible to set up a dedicated sediment module for this task. We lack detailed habitat mapping for the study area, and there are no reliable SPM flux measurements at the open boundary. In addition, the specific properties of the suspended particles in the region remain uncertain. The available SPM observational data reflect total concentrations, combining both organic and inorganic materials without differentiating between particle types. While the NN performs well in predicting SPM, it does not explain why SPM dynamics differ between the two stations, though the difference in depth is a contributing factor. This is where Lagrangian simulations provide valuable insight. They illustrate the delayed wind response at the deep station compared to the shallow site. This is a pattern that holds across different wind scenarios.

**In the numerical modelling part, only tide forcing is considered because of its dominate role in driving the currents in the bay. However, wind is the key factor to be studied in**

**this work – the title of this manuscript starts with "wind". Why not include winds in the numerical modelling? Note that the so called tidally induced transport time scale (133 hours) is also comparable to the synoptic weather time scales (several days).**

When we consider wind stress we account for different mechanisms of its influence on SPM dynamics, as wind stress determines shear in the water column and mediates wave-breaking processes. Tidal forcing, of course, also induces resuspension and the information about tidal phase and elevation is included in NN as well. However, in addition, tides strongly contribute to net transport (particles do not return to their original location after a tidal cycle) or advection of material within the domain due to its shallowness, complex topography and large tidal amplitudes (all non-linear terms - bottom friction, advection of momentum and non-linearity in the continuity equation- play significant role therefore such a strong net transport exists). Our Lagrangian simulations showed this tidally induced transport. Of course tidally induced transport can be modified by the winds; however, the tidally induced transport works permanently in time. To support this, we conducted additional simulations under moderate wind conditions, which showed that the main transport pathways were preserved, while wind forcing mainly increased dispersion (Konyssova et al., 2025).

**Numerical model validation is only on tides using tide gauge data. However, the currents are not validated at all. The latter is relevant to the transport pathways. Note that the dynamical mechanisms are different. Accurate tides do not guarantee correct currents.**

The model has been utilized in many studies (Fofonova et al., 2019; Kuznetsov et al., 2020, 2024; Neder et al., 2022; Sprong et al., 2020; Sidorenko et al., 2025) and has shown the ability to reproduce the tidal dynamics effectively. This same setup for the Sylt-Rømø Bight has also been implemented in Fofonova et al. (2019), where both sea surface heights and current velocities were validated.

**The "passive tracers" used in the numerical model experiments in this work are actually Lagrangian particles as commonly referred to in the coastal ocean modelling community. These should be called "virtual particles", "Lagrangian particles", or simply "particles" in Lagrangian particle tracking modelling. In contrast, "tracers" (or "passive tracers") are often used in numerical dye experiments, which is associated with initial concentrations when they are released in the numerical model and their concentration change in space and time. It is better to change "tracers" to "particles" to avoid confusion. Otherwise, it is important to point out the tracers used in this study are different from other tracer model applications.**

In the manuscript, we used the term "passive tracers" instead of "particles" to specify that the tracers (or particles) we released in our model are passive and do not have weight. We appreciate the reviewer's point that in some contexts, "tracers" may be associated with

concentration-based dye experiments. To avoid potential confusion, we have added a clarification in the revised manuscript that our use of "tracers" refers to Lagrangian particles and is different from "dye" experiments.

**Linear correlation is heavily used in the analysis (Figures 6 and 7). However, it is not clear at all whether a linear relationship exists between the two variables (SPM and wind speed). Actually, the scattered data points indicate a non-linear relationship.**

While linear correlation offers a first-order diagnostic, we acknowledge that it may not fully capture the complex, non-linear relationships between wind and SPM concentrations. To better address this, we employed a Neural Network model, which is capable of capturing such non-linear interactions, as discussed in Section 3.3.

**There are some awkward sentences and wrong word choices. For example, L20, "this study integrate ..." should be changed to "in this study, we integrate ...".**

We rephrased the sentence following the suggestion.

**Some of the statements are not rigorous at all: Lines 269-270, "... patterns reflecting the combined influence of biological activity and meteorological forcing" – how do you know that?**

We acknowledge the reviewer's comment and have revised the sentence to be more cautious, referring to the forthcoming analysis rather than asserting conclusions upfront.

......................................................................................................................................................

References:

Fofonova, V., Androsov, A., Sander, L., Kuznetsov, I., Amorim, F., Hass, H. C., and Wiltshire, K. H.: Non-linear aspects of the tidal dynamics in the Sylt-Rømø Bight, south-eastern North Sea, Ocean Sci., 15, 1761–1782, https://doi.org/10.5194/os-15-1761-2019, 2019.

Konyssova, G., Sidorenko, V., Androsov, A., Sander, L., Danilov, S., Rubinetti, S., Burchard, H., Winter, C., Wiltshire, K.H.: Changes in tidal dynamics in response to sea level rise in the Sylt-Rømø Bight (Wadden Sea), Ocean Dynamics, 75, 43, https://doi.org/10.1007/s10236-025-01688-1, 2025.

Kuznetsov, I., Androsov, A., Fofonova, V., Danilov, S., Rakowsky, N., Harig, S., and Wiltshire, K. H.: Evaluation and Application of Newly Designed Finite Volume Coastal Model FESOM-C, Effect of Variable Resolution in the Southeastern North Sea, Water, 12, 1412, https://doi.org/10.3390/w12051412, 2020.

Kuznetsov, I., Rabe, B., Androsov, A., Fang, Y.-C., Hoppmann, M., Quintanilla-Zurita, A., Harig, S., Tippenhauer, S., Schulz, K., Mohrholz, V., Fer, I., Fofonova, V., and Janout, M.: Dynamical reconstruction of the upper-ocean state in the central Arctic during the winter period of the MOSAiC expedition, Ocean Sci., 20, 759–777, https://doi.org/10.5194/os-20-759-2024, 2024.

Neder, C., Fofonova, V., Androsov, A., Kuznetsov, I., Abele, D., Falk, U., Schloss, I. R., Sahade, R., and Jerosch, K.: Modelling suspended particulate matter dynamics at an Antarctic fjord impacted by glacier melt, Journal of Marine Systems, 231, 103734, https://doi.org/10.1016/j.jmarsys.2022.103734, 2022.

Rubinetti, S., Kuznetsov, I., Fofonova, V., Androsov, A., Gnesotto, M., Rubino, A., and Zanchettin, D.: Water Mass Transport Changes through the Venice Lagoon Inlets from Projected Sea-Level Changes under a Climate Warming Scenario. Water 15, 3221, https://doi.org/10.3390/w15183221, 2023.

Sidorenko, V., Rubinetti, S., Akimova, A., Pogoda, B., Androsov, A., Beng, K. C., Sell, A. F., Pineda-Metz, S. E. A., Wegner, K. M., Brand, S. C., Shama, L. N. S., Wollschläger, J., Klemm, K., Rahdarian, A., Winter, C., Badewien, T., Kuznetsov, I., Herrling, G., Laakmann, S., and Wiltshire, K. H.: Connectivity and larval drift across marine protected areas in the German bight, North Sea: Necessity of stepping stones, Journal of Sea Research, 204, 102563, https://doi.org/10.1016/j.seares.2025.102563, 2025.

Sprong, P. A. A., Fofonova, V., Wiltshire, K. H., Neuhaus, S., Ludwichowski, K. U., Käse, L., Androsov, A., and Metfies, K.: Spatial dynamics of eukaryotic microbial communities in the German Bight, Journal of Sea Research, 163, 101914, https://doi.org/10.1016/j.seares.2020.101914, 2020.

---

## Author Comment (AC2)

**Response to Reviewer #2**

**Manuscript title:** *"Wind and Phytoplankton Dynamics Drive Seasonal and Short-Term Variability of Suspended Matter in a Tidal Basin"*

We thank Reviewer #2 for the thoughtful and thorough evaluation of our manuscript. We value the recognition of our combined methodological approach and appreciate the constructive comments provided. In the following, we respond to each comment in detail and explain the revisions carried out in the manuscript.

Reviewer comments are shown in **bold**, and our responses are given in regular font below each comment.

..............................................................................................................................................................

Review of the manuscript "Wind and Phytoplankton Dynamics Drive Seasonal and Short-Term Variability of Suspended Matter in a Tidal Basin" by Konyssova et al., submitted to Biogeosciences (egusphere-2025-2135).

**The manuscript is well written (though a bit repetitive in places) and has many supporting graphs. The study is essentially very local but has useful conclusions for similar systems, at least throughout the Wadden Sea. The main conclusions are not new but quantify the separate contributions from the separate drivers for this particular inlet during winter and summer. The approach is novel, combining observations, particle tacking and a neural network (NN) model to identify changes in (usually) already weak corelations. Personally I found the particle tracking study the most interesting part, and a good way to identify source regions without doing a full morphodynamic study (and truly considering SPM dynamics). The manuscript lacks some information in places (like the predictors included in the NN), seems to contain some errors in listing results, does not always include enough evidence for statements, does not specifically reference the supplementary materials and relies in many parts on weak and even weaker correlations. The latter I consider a suboptimal bid for information, but as sediment resuspension is a notoriously difficult process to predict given the large range of spatial and temporal scales involved I do find it understandable and, to be fair, a brave choice. Overall the authors make a relatively convincing case for the identified drivers and their seasonal impact, though they should be cautious with their conclusions when considering observational correlations and biological proxies. They are however much more cautious with these results in their discussion and conclusions. More detailed comments are listed below.**

**Recommendation**

**Major revision, though new simulations are not necessary.**

We thank the reviewer for their thorough and constructive evaluation of the manuscript. We appreciate the recognition of our combined methodological approach and the potential relevance of our findings to similar coastal systems.

In response to the reviewer's detailed comments, we have revised the manuscript to enhance the clarity and focus of the storyline, remove repetitive text and overstated claims. Some supporting figures have been moved to the Supplementary Material, and we have improved cross-referencing throughout the text to guide readers more precisely to the relevant figures and tables. We also appreciate the reviewer's point about the difficulty of resolving SPM dynamics across scales and have taken care to present our conclusions with appropriate caution.

**Detailed Comments**

1. **Line 56: I get the point but the wording is awkward, suggesting there is a sediment balance to begin with.**

   We have rephrased the sentence to avoid confusion and misinterpretation. We see how the original wording may suggest that a defined sediment balance already exists.

2. **Section 2.1: in the area description I miss mention of possible anthropogenic activity in the region that could influence observed SPM levels. Is there local ship traffic (commercial, recreational), ferry service, dredging or trawling activity?**

   Thank you for this important remark. Indeed, anthropogenic activities in the Wadden Sea such as maritime traffic, dredging, and trawling have been shown to influence sediment dynamics and turbidity. Unfortunately, there's no specific data available on these activities in the research area to include directly in our analysis. However, we have now added a brief contextual disclaimer in the Area Description section, noting that such activities may contribute to local SPM variability and should be considered in future studies.

3. **Line 147: I am a little bit surprised that M6 is not included here, but have not checked the listed references for its importance.**

   The M6 constituent is not included when generating elevation at the open boundary, which is situated largely in a relatively deep area of the domain, as its magnitude is much smaller than that of the M4 overharmonic, which is included. However, M6 is, of course, generated within the domain.

4. **Line 153: please be table or figure specific when referring to the supplementary materials. This applies to the whole manuscript.**

   Thank you for this helpful suggestion. We have reviewed the manuscript and revised all references to the supplementary materials to explicitly mention the corresponding table or figure numbers where applicable.

5. **Line 161: needs rephrasing, e.g. "calculated ... were released" or 'We released ...'**

   We rephrased the sentence for clarity and grammatical correctness.

6. **Line 233: here winter is mentioned for the first time outside of the abstract and introduction, so I would expect a definition of the seasons used here. Note that different research fields can have different delineations of the year into seasons. The Wadden Sea Quality Status report of 2017 even used January-March for winter (van Beusekom, 2017), and astrological definitions do not follow whole months. So please specify which delineation was used in this work.**

   **Reference: van Beusekom, J.E.E., Bot, P., Carstensen, J., Grage, A., Kolbe, K., Lenhart, H.-J., Patsch, J., Petenati, T., Rick, J. (2017), Eutrophication, in the Wadden Sea Quality status report, version 1.01 (https://qsr.waddensea-worldheritage.org/reports/eutrophication)**

   We appreciate this critical remark. This nuance has been missed in the original version and now has been clarified explicitly in the revised text. Since the neural network analysis aimed to separate periods of minimal biological activity in winter from spring bloom, summer, and autumn, we defined the seasons based on observed seasonal cycles in Chl-a concentrations at the study site rather than using calendar-based or astronomical definitions. Specifically, we used the following delineations:
   - Winter: November 20 – February 19 (low biological activity, low Chl-a)
   - Spring: February 20 – May 31 (phytoplankton bloom initiation and peak)
   - Summer: June 1 – September 19 (post-bloom conditions, high light, reduced Chl-a)
   - Autumn: September 20 – November 20 (transitional period).

7. **Line 239: I'm not sure what is meant here with "current elevation", but I assume the authors mean SSH? And a table overview of the predictors of the NN model is necessary, likely in the supplementary material. Counting the listed predictors here I find 5 directly mentioned predictors or 19 when counting the individual lag components separately. But not the 21 mentioned in 3.3.3 for the non-biotic**

**version of the NN. Please list the basic and added (biological proxy) NN model predictors clearly.**

Yes, by "current elevation" we indeed meant sea surface height (SSH), and we have corrected the terminology in the revised manuscript for clarity. We have now also included a table in the Supplementary Material that explicitly provides an overview of all input variables used in both the abiotic and full (biotic + abiotic) NN models.

8. **Figs 4, 6, 7: these could go into the supplementary materials with simple a table in the main text listing the correlation metrics in one table for all figures. This would facilitate comparison of the individual month correlations over the different observation drivers with observed SPM.**

These figures illustrate not only the correlation between parameters but also the variability and scatter structure, which are important for interpreting the seasonal character of the relationships. While the text highlights some key quantities, we have now added a table in the Supplementary Material summarizing the monthly correlation metrics, following the reviewer's suggestion.

9. **Line 250: I miss references here to more work regarding EPS, its production by phytoplankton and its role in flocculation.**

We have added a reference to support the role of EPS production in flocculation. As the paragraph is primarily descriptive, we have moved it to the Introduction in response to the suggestion of Reviewer #1.

10. **Line 274: the data also shows an autumn bloom which seems to have a different impact on SPM dynamics compared to the spring bloom mentioned here. Some explanatory text on this phenomenon (why is there an autumn bloom in this shallow inlet and why does SPM increase at the same time? ) would be useful here.**

In the revised manuscript, we have moved the paragraph originally located at L270–276 to the Introduction, following a suggestion by Reviewer 1. Nonetheless, we agree that this point deserves attention, and we now address this in the Discussion section by highlighting the seasonal differences in biological activity and its interaction with physical drivers such as wind. Elevated SPM concentrations are observed during secondary phytoplankton bloom in autumn, likely due to the combined effects of stronger wind conditions promoting the resuspension process and organic matter availability to sustain the secondary phytoplankton bloom. This explanation is supported by the NN results (Table 2, Fig. 14), which show that during autumn,

predictions overestimate observed SPM, suggesting additional biological regulation at play.

11. **Line 291: the use of "*winter (December-January) ... autumn (August-October)*" suggest a seasonal definition that has not been specified before and which suggest unequal length of the seasonal delineation. And which is at odds with other parts of the manuscript (e.g. line 299 where winter is suddenly December-February, while in line 312 late autumn consists of November and December but in line 328 it is listed as October-November).**

We acknowledge the inconsistency in how seasonal terms were used in the manuscript. In our neural network, we employed seasonality based on observed Chl-a seasonal cycles, as mentioned in our response to comment 6 (L233). We have revised the manuscript to avoid the inconsistent use of the terminology and removed any ambiguous references, such as "late autumn", instead referring directly to the corresponding months.

12. **Line 302: the listed do not match those in fig. 4 where December has 0.84 at the shallow station and January has 0.75 at the shallow station.**

We thank the reviewer for noting the mismatch. The initial manuscript incorrectly attributed the highest correlation in January ($R^2 = 0.84$) to the deep station, while it is observed at the shallow station. We have corrected this in the revised text and now clearly state that the highest correlations occur in December-January: $R^2 = 0.84$ and $R^2 = 0.75$ at the shallow station, and $R^2 = 0.52$ and $R^2 = 0.56$ at the deep station.

13. **Line 306: all $R^2$ are < 1 as $R^2=1$ would be a perfect fit. So not sure why this is included here.**

We agree that this statement was odd and have removed it from the manuscript.

14. **Line 309: please include a list of predictors used in each separate NN model application.**

We appreciate the reviewer's suggestion. We have revised the manuscript to clarify that "current elevation" refers to sea surface height (SSH) and now provide a complete overview of the NN model predictors in a table, including both the abiotic and extended biological models. This also clarifies the total number of inputs used in each case.

15. **Line 339: $R^2=0.44$ is by no means a strong correlation and it is not even the highest correlation found as April lists $R^2=0.50$ for the shallow stations correlation between wind speed and SPM levels. And it should be mentioned**

**clearly in the text that a correlation between 2 timeseries is by no means a causal relationship, as demonstrated by the strong correlation between winter Chl-a and SPM levels. Therefore I find line 341 rather contentious: it was simply tested here if there was a correlation between 2 timeseries, and for January only 44% of the SPM variability was explained by the instantaneous wind speed and in April 50%. Hardly a dominant control in winter I think. In my opinion the authors should be more careful in their statements regarding the observational time series correlations.**

Following the suggestion, we have corrected this overstatement in the revised text and adjusted the wording to more accurately reflect the strength of the correlations without implying causation. Rather than suggesting a dominant control, we now interpret the seasonal pattern in wind–SPM correlation more cautiously, emphasizing that wind forcing appears to contribute more substantially to SPM variability during winter than in other seasons.

16. **Line 353: "*The stronger relationship*" is more accurately a less weak relationship.**

   We agree that "stronger" may be misleading and have revised the sentence to reflect a more accurate interpretation of the correlation strength.

17. **Line 357: I would say that wind plays a role in controlling SPM variability on different temporal scales, but a crucial role suggests higher correlation factors than have been found here. Besides, the authors only consider tides in the particle tracking study because they are the dominant resuspension mechanism (line 428), and do not include wind waves here.**

   The relatively low correlations can be explained by the non-linear nature of the wind's impact on SPM dynamics, which is, however, captured by the neural network. Indeed, when we include wind stress in the neural network, we account for different mechanisms of its influence on SPM dynamics, as wind stress determines shear in the water column and mediates wave-breaking processes. Tidal forcing, of course, also induces resuspension and the information about tidal phase and elevation is included in NN as well. In addition, tides strongly contribute to net transport (particles do not return to their original location after a tidal cycle) or advection of material within the domain due to its shallowness, complex topography and large tidal amplitudes (all non-linear terms - bottom friction, advection of momentum and non-linearity in the continuity equation- play significant role therefore such a strong net transport exists). Our Lagrangian simulations show indeed tidally induced transport but not mixing. The sentence has been revised to avoid overstatement and more accurately reflect the wind's role in modulating SPM variability.

18. **Line 373: I'm sorry but I don't see this in Figure 8 (rather the opposite) and any kind of metrics to support this statement are missing. Please include more supporting evidence here.**

We thank the reviewer for this observation. The visual difference between low and high tide distributions in Fig. 8 may appear subtle, especially when considering the total number of data points. However, the distinction becomes more evident when examining the upper envelope of the data distribution. At both stations, particularly the shallow one, the highest observed SPM concentrations occur during low tide. This pattern is consistent with the expectation that, at lower water levels, the sample intake is closer to the sediment surface and thus more likely to capture freshly resuspended material. While we recognize that this is a secondary aspect of the study, we retained this analysis in the Supplementary Materials.

19. **Line 515: a reference to Table 2 is missing here as a source for the 80% reduction mentioned.**

Revised accordingly.

20. **Lines 518-520: there is no reference here to figure S1 and the supplement does not contain any metrics for a direct comparison of seasonal wind speed. Indeed, autumn winds look a lot stronger in S1 than the spring or summer winds. The figure quality of S1 is too low as well. But I see no evidence for the statement in these lines.**

We thank the reviewer for pointing this out. We have replaced the original low-quality Figure S1 with a higher-resolution version. We also clarified that Figure S1 presents wind directions (wind rose) rather than wind speeds. The wind speed seasonality has been described in the dedicated section "Role of wind forcing in seasonal SPM variations."
Regarding the explanation, we revised the text to clarify that the discrepancy between model predictions and observations cannot be explained by wind forcing alone, particularly in biologically active seasons. The abiotic-only model systematically overestimates SPM in spring and summer, highlighting the increasing role of biological processes, while better correlations in autumn reflect conditions more similar to winter, when the model was trained.

21. **Lines 520-522: or it speaks to the stronger winds in autumn?**

Thank you for the comment. The text now clarifies that differences in correlation between spring and autumn are interpreted in the context of the NN trained on winter abiotic conditions. The lower correlation in spring is attributed to stronger biological

influences that are not represented in the abiotic-trained model. In contrast, autumn conditions, with lower biological influence and similarly strong wind conditions, more closely resemble the physical environment in winter.

22. **Line 523: tidal dynamics was already mentioned as the dominant driver of SPM conditions, now biological processes are mentioned here as being dominant in summer. I would suggest that biological processes become the primary driver of SPM variability in summer, not of SPM dynamics.**

Yes, you are absolutely right. The sentence has been rephrased to clarify that biological processes are considered the primary driver of SPM *variability* during summer.

23. **Section 3.3.3: it would be good to include seasonal plots for the 2 additional NN model applications (all seasons physics and all seasons plus biological proxies) in the supplementary materials. That is the equivalent of figure 14.**

We appreciate this suggestion that seasonal plots for the two extended NN models would improve the interpretability and allow more precise comparison with the baseline winter-trained model. We have now added tables with seasonal predicted values and figures with regression plots for both the all-seasons physical-only and the all-seasons physical + biological proxy models to the Supplementary Material:

[Figure]

Upper row: Regression analysis for NN trained on all seasons' abiotic data and applied to spring, summer and autumn data vs observations.
Lower row: Regression analysis for NN trained on all seasons' abiotic data *with biological proxies* and applied to spring, summer and autumn data vs observations.

**24. Section 3.3.4: the title is not semantically correct.**

Thank you for pointing this out. We have revised the section title to "Neural network trained on all-season data including biological proxy features".

**25. Line 542: please reference figure S2 directly.**

Revised accordingly.

**26. Lines 553-554: Amen to that. I think the statement here is more accurate than other statements throughout the manuscript mentioning crucial or dominant drivers. And the authors should not sell themselves short: this is not new in itself but they have helped quantify the parts played by different drivers for this inlet and for these data years.**

We thank the reviewer for this supportive and thoughtful remark! In line with this suggestion, we have revised some of the earlier sections in the manuscript to avoid overstatements regarding "dominant" or "crucial" drivers. We now more consistently emphasize the variable and seasonally dependent *roles* of different processes, while highlighting the study's contribution in quantifying their relative influence rather than attempting to resolve all processes involved.

**27. Section 4.1: I miss a more detailed discussion about the biologically induced flocculation process here, as for instance TEP (transparent exopolymer particles) production occurs under certain circumstances and therefore cannot be predicted by Chl-a levels.**

The discussion in Section 4.1 has been expanded to include the role of transparent exopolymer particles (TEPs) in biologically mediated flocculation. We note that TEP production can be influenced by factors beyond phytoplankton biomass, such as nutrient availability, species composition, and abiotic conditions, adding complexity to the relationship between SPM and Chl-a. This perspective puts our findings in a broader context while maintaining the interpretation that Chl-a patterns provide meaningful insight into seasonal SPM variability.

**28. Line 571: I miss a reference here to Hommersom et al (2009) who report a correlation between SPM and Chla due to resuspension in their work. A discussion of this work and the difference with the present study should be included.**

**Reference: Hommersom, A., Peters, S., Wernand, M. R., & de Boer, J. (2009). Spatial and temporal variability in bio-optical properties of the Wadden Sea. Estuarine, Coastal and Shelf Science, 83(3), 360-370.**

A reference to Hommersom et al. (2009) has been added to expand the discussion with their findings of a strong Chl-a–SPM correlation caused by resuspension that are consistent with our winter observations.

29. **Line 590: "*constrained*", do you mean unknown?**

Yes, with "constrained" we meant that the role of introduced species is not well known in the study site. We agree that the original phrasing is ambiguous and have rephrased it to "remains less clear".

---

## Author Comment (AC3)

**Response to Reviewer #1**

**Manuscript title:** *"Wind and Phytoplankton Dynamics Drive Seasonal and Short-Term Variability of Suspended Matter in a Tidal Basin"*

We would like to sincerely thank Reviewer #1 for their detailed and constructive feedback. We greatly appreciate the time and effort invested in reviewing our manuscript. Below, we provide a point-by-point response to each of the comments and describe the corresponding revisions made to the manuscript.

Reviewer comments are shown in **bold**, and our responses are given in regular font below each comment.

....................................................................................................................................................

**The authors introduced an interesting approach to investigate the effect of wind and phytoplankton on the seasonality in SPM concentration. Although the subject is of great interest, the study itself has many flaws and shortcomings. The manuscript is too long, with many repetitions, there is no clear difference between results and discussions. The flaws are**

**1) The SPM concentration time series is patchy and does not catch tidal variability. This may lead to an overestimation of wind effects as tidal variability is not included in the data.**

The SPM concentration time series is patchy due to the nature of field-based sampling. Nevertheless, the dataset includes the exact times of the sampling, which allows us to identify the moment of the tidal phase accurately. We have reconstructed sea surface height (SSH) and tidal phase for all sampling time instants for the considered years (2000-2019) using harmonic analysis of model output with a resolution of 5 minutes. These reconstructed tidal variables were included as features in the statistical and neural network analyses to help account for tidal influence at the time of each sample. In addition to the tides, the wind conditions were retrieved from the meteorological station based on the List (Sylt/ Germany) for the whole duration of the time-series. We would like to note that the length of the time series spanning over 20 years provides statistically robust coverage across a range of tidal phases and seasonal conditions.

**2) The Lagrangian particle trajectories are not representative for SPM transport. The added value of the model results is limited.**

We agree that Lagrangian tracking cannot replace a sediment module, and we did not claim otherwise. There are several reasons why we chose not to run a full sediment module and instead used Neural Network (NN) and Lagrangian simulations.
First, we lack detailed habitat mapping for the study area, and we do not have reliable suspended particulate matter (SPM) fluxes at the open boundary. Additionally, the specific properties of the particles relevant to this region remain uncertain. The available SPM observational data represent total concentrations, encompassing both organic and inorganic

materials, without distinguishing between different particle types. Given the relatively small spatial scale of the study area, such uncertainties are particularly significant and limit the effectiveness of traditional sediment modeling approaches. Furthermore, our study spans a 20-year period, during which numerous potential forcing scenarios exist. For this reason, we opted to use a Neural Network trained on observational data to predict SPM concentrations and to assess the role of biological activity at two sampling locations. But the NN does not explain why the SPM dynamics differ between the two stations, apart from the evident influence of depth. While we acknowledge that passive tracers do not capture the full complexity of SPM transport (e.g., settling, flocculation, or bed interaction), the purpose of this component was not to simulate sediment dynamics directly. Rather, it aimed to explore hydrodynamic connectivity and dominant transport pathways within the basin. As also positively noted by Reviewer #2, the passive tracer simulations provide an efficient and computationally inexpensive means of identifying likely source regions and transport timescales, without requiring detailed sedimentological input, which is often highly uncertain. The simulations, run over two synodic months, capture statistically robust tidal transport patterns. These results help explain, for example, the delayed wind response observed at the deep station compared to the shallow site, a pattern confirmed across different wind scenarios.

We have clarified the scope and limitations of the particle tracking experiment in the revised manuscript, emphasizing that these trajectories are not intended to represent particulate fate but to provide insight into spatial connectivity and transit times under tidal forcing.

**3) As the correlation between wind and SPM concentration is low, the authors still ascribe the higher winter concentrations (when biological effects are small) to claim wind effects. Do you need wind to explain the seasonality? Would biology be not sufficient. What are the arguments to claim that higher SPM concentrations align with winds?**

We acknowledge that the correlation between wind speed and SPM concentration is not particularly strong. However, this is expected given the inherent complexity and non-linearity of coastal dynamics. Nevertheless, wind remains an important factor in explaining the observed seasonal patterns in SPM concentrations.

The Sylt-Rømø Bight is a relatively shallow area, where wind stress and tidal forcing induce velocity shear and generate significant turbulence throughout the water column, including near the bottom. A typical storm event in the North Sea can mix the water column to depths of up to 50 meters within just two days (see the FLEX experiment in Burchard et al., 2002; Notably, the modeling results from that study, which did not include wave breaking, were still able to reproduce this mixing efficiently). This turbulence is a primary driver of sediment resuspension in shallow areas (e.g., Fettweis et al., 2012; Stanev et al., 2009). Exactly for this reason, the Neural Network trained using only abiotic conditions was able to predict winter-season SPM concentrations with high accuracy.

While biological activity does not directly cause resuspension or generate currents, it can influence resuspension processes by modifying the properties of the bed substrate and affecting flocculation dynamics. The key question is to what extent biological activity also contributes to the overall variability of SPM. While biological factors such as flocculation and

sediment stabilization are central to the spring-summer decline in SPM concentrations, they are insufficient to explain the consistently higher concentrations observed during winter months.

**4) The effect of biological processes on SPM dynamics is described indirectly using Chl-a and temperature as a proxy of phytoplankton. This is not convincing as both parameters do not reflect the complicated biological processes.**

We agree that the biological processes influencing SPM dynamics are complex and cannot be fully captured by chlorophyll-a and temperature alone. We are aware that flocculation, grazing, benthic–pelagic coupling, and biofilm formation all depend on species composition, microbial activity, and organic matter characteristics. Such data that could help us separate between grazing, benthopelagic coupling, flocculation, etc., is unfortunately not available. Nevertheless, chlorophyll-a remains a broadly used proxy for phytoplankton biomass and, by extension, biological influence on particle aggregation (e.g., Cloern, 1999).
We have clarified these limitations and now emphasize that our interpretation is exploratory rather than conclusive. The Neural Network results are discussed in the context of these constraints, and we explicitly acknowledge that more targeted biological data (e.g., TEP, EPS, biofilm formation) would be needed to fully resolve biological contributions to SPM dynamics.

**5) Physical and biological induced flocculation is not well described in the manuscript, resulting in statements that are only partially true.**

We appreciate the reviewer's attention to the details of flocculation processes. However, we would like to clarify that this study is not focused on flocculation per se, but rather on the broader set of physical and biological processes influencing SPM concentrations. We acknowledge flocculation as one of the most direct biologically mediated mechanisms contributing to SPM variability, but we do not attempt to resolve it mechanistically. Moreover, our dataset does not allow for the separation of individual biological processes involved, including not only flocculation, but also grazing, benthic–pelagic coupling.
In response to the reviewer's feedback, we have revised the Introduction to more accurately describe flocculation and its biological and physical drivers, while maintaining the overall focus on integrated physical–biological controls of SPM concentrations.

**SPM, SPM level etc. is not a synonymous of SPM concentration: when you mean the latter add (replace by) 'concentration'.**

Revised accordingly.

*1. Introduction*

**The introduction needs to be rewritten as it is a now a string of sentences without a common storyline. It should remind the state of the art background information (many references are from older literature >25yrs, which is ok, but there has been some progress since) and sets out the main research questions and why they are of importance.**

We appreciate the reviewer for the feedback. We agree and have revised the manuscript referring to a more recent literature and tightened up our Introduction.

**L43: What do you mean by 'primary production gradients'**

In the revised manuscript, we clarified that the term refers to spatial variability in primary productivity that is associated with light limitation caused by suspended particulate matter.

**L45-46: I would not introduce here the study site.**

Following the suggestion, we have removed the early introduction of the Sylt-Rømø Bight from this paragraph. However, we retained the broader context of Wadden Sea-specific processes, as they provide important background for the mechanisms discussed in the study.

**L49: Not clear how the import of SPM and OM is reflected by the heterotropic nature of the Wadden Sea. Import means from the North Sea?**

We appreciate this comment and agree that the sentence lacked clarity. We have therefore removed the sentence from the revised manuscript.

**L51: Above you describe the Wadden Sea as a tidal basin, here you switch to an intertidal area. The reference in L53-55 are not about intertidal areas. I suggest to not focus here on intertidal areas**

To maintain consistency and avoid confusion, we have revised the sentence and removed the specific focus on intertidal areas. However, as the mechanisms remain relevant, we retained the description to refer more generally to sediment accumulation in the basin.

**L62: Schubel, 1974 does not seem the right reference (it is about sediments and not wind effect)**

This reference was originally included to provide historical context on how storms can impact sediment dynamics, but we agree it is not directly relevant for describing wind-induced resuspension in the Wadden Sea. We therefore removed it here to keep the introduction focused on more directly applicable studies.

**L63: Fettweis et al. (2012) is not about flocculation**

This reference was originally included to illustrate meteorological variability of suspended matter in the North Sea, but its placement in the flocculation context could have been misleading. We have therefore relocated it to the sentence focusing on weather forcing of SPM dynamics instead of flocculation.

**L65: This is a repletion of L41**

We revised the paragraph to avoid repetition and clarified the role of biochemical processes without duplicating the earlier text.

**L68: Fettweis & Van den Eynde is not about biological processes**

We have removed this reference from the sentence.

**L68-69: Flocculation is a process of aggregation and break up. It is enhanced by the occurrence of marine gels (TEP), but occurs also without biological influence, due to the cohesive properties of fine-grained minerals (especially clay minerals).**

Thank you for pointing out this nuance, we agree that it was not clearly stated. The revised text now clarifies that flocculation involves both aggregation and breakup and can occur due to both biological (e.g. TEPs) and physical (e.g. clay cohesion) mechanisms.

**L70-72: Flocs nearly always contain OM in various forms (adsorbed OM molecules, detritus and living OM).**

Thank you for this helpful detail, we agree. We have revised the sentence to clarify that natural flocs contain not only detritus but also organic matter in various forms.

**L72-74: Add a reference**

A reference has been added to support the role of floc characteristics in governing settling velocities and resuspension thresholds.

**L80: Not clear how 'excessive nutrient loads promote organic sedimentation'. Replace 'organic' by 'organic matter'. How can organic matter (generally a density close to water density) settling without mineral ballast (biomineral floc)? Add a reference.**

We have revised the text to clarify that excessive nutrient loads promote OM production, which can then form biomineral flocs and enhance sedimentation.

**L83: How important is this (that can impact seagrass) in your study?**

We agree that this reference to seagrass may imply a broader ecosystem focus than is within the scope of our study. We have revised the sentence to better reflect the relevance of fluctuating turbidity in shallow benthic environments, without overextending to specific ecological impacts.

**L85-86: It is becoming a fast evolving research topic, and many new research has been published on this subject.**

Thank you for the comment. We rephrased the sentence to reflect the growing interest in this research direction and updated the manuscript to reference relevant recent studies where appropriate.

**L96: Two monitoring station, above you only mention one.**

In this sentence, we were referring specifically to the meteorological data, which comes from a single weather station in List (Sylt, Germany). The two LTER stations mentioned earlier refer to the *Sylt Roads* monitoring sites. To avoid confusion, we have clarified the wording in the revised manuscript to distinguish between the meteorological station and the two biogeochemical monitoring stations.

*2. Data and Methods*

**L110-112: What are the 10-20% other than tidal forcings?**

According to Fofonova et al. (2019), more than 80% of depth-averaged velocity variability in the Sylt-Rømø Bight is explained by tidal forcing, while the remaining 10–20% can be attributed to the wind stress and density gradients (baroclinic effects).

**L119: The two references are not on tides. They are probably not needed here. Here the tidal range is 1.7m while on L109 it is 2m.**

The value of 2 m in L109 refers to the average entire Sylt-Rømø Bight, while L119 refers specifically to the Königshafen embayment, which is more sheltered and exhibits slightly lower tidal amplitudes. However, to avoid confusion and since the cited references are not directly focused on tidal characteristics, we have removed the sentence and the associated references in the revised manuscript.

**L144-145: I don't think that higher harmonics and over-harmonics need references. Rephrase L144-149.**

We removed the references to avoid over-citation and rephrased the sentence to keep it concise.

**L160: You released tracers in the intertidal areas, correct? At the beginning of inundation?**

Tracers were released from all grid cells that are consistently inundated during flood phases. Still, the releases occurred at fixed three-hour intervals, allowing us to capture a full range of tidal phases and ensure statistical robustness. This was explained initially on L175–176, but we moved the justification sentence to L164 for better coherence.

**L172-174: How can the passive tracers be representative for SPM, that undergoes resuspension and deposition phases?**

We recognize that passive tracers cannot capture the full complexity of SPM behavior, particularly resuspension and deposition dynamics. However, our goal in using Lagrangian simulations was not to model sediment fate per se, but to assess hydrodynamic connectivity and typical transport timescales within the basin. These simulations, while offering an efficient and computationally inexpensive approach, help us explain patterns such as the delayed wind response at the deep station that is not so clear otherwise. (We provide a more detailed explanation of this approach in our response to point (2) in the general comments).

**L187-188: Can you mention the sampling frequency and the total amount of data. Was the sampling done during each weather conditions? Are the samples taken at the surface?**

The sampling frequency is twice a week at 1 m below the sea surface, as mentioned on L192-193 of the original manuscript. Regarding weather conditions, the measurements have not been taken in extreme storm conditions, depending on the weather/sea state restrictions of the ship operations. More details of the dataset description and evaluation are provided by Rick et al. (2017; 2023).

**L190-192: Be a little more specific with the method. Do you use HPLC for Chl?**

HPLC is not used for chlorophyll-a analysis. The chlorophyll-a concentrations were determined spectrophotometrically following the trichromatic method after 90% acetone extraction, as described in established protocols (UNESCO, 1966; Jeffrey & Humphrey, 1975). To be specific, we add this clarification in the revised manuscript.

**L195-196: The reference to PANGAEA is not useful, as there you have to search for the right files. I have downloaded the data in Rick et al. (2023), they contain data from to the period 2014-2019. The data before 2014 can be found in Rick et al. (2017) https://doi.org/10.1594/PANGAEA.150032**

**Can you check if these cover all the data you have used.**

Thank you for your comment and for taking the time to review the data behind the study. We referred to the survey by Rick et al, 2023, not to list all 19 links of the Pangaea. Their study does provide dataset links for data from 1973 to 2013, and separately for each year from 2014 to 2019 in their abstract. As we focused on the relatively consistent de-eutrophication period 2000-2019, the dataset provided and discussed in Rick et al, 2023, and the Pangaea links therein cover all the LTER data we have used in our study.
In response to this comment, we added all the links in the Data Availability.

**L200-203: The link can be omitted as also available in 'Data Availibilty'**

The link has been removed from the main text, as suggested.

**L205: Sea surface elevation versus SSH : use height instead of elevation?**

Although "sea surface height" and "sea surface elevation" are often used interchangeably, we agree that using "height" is more consistent with the abbreviation "SSH" and have revised the manuscript accordingly.

**L210: Do you mean; when the samples (cfr PANGAEA) were taken?**

Yes, we refer to the exact timestamps of the LTER samples (Sylt Roads dataset). As tidal measurements were not included in the original dataset, we reconstructed the SSH signal at the sampling locations using the validated model output. This allowed us to estimate tidal phase and elevation gradients at the time of sampling. The manuscript has been clarified accordingly.

**L215: Can you specify which environmental parameters were used to predict SPM concentration? How do they relate to SPM concentration?**

The predictors used in the NN model are described in detail in the manuscript (in the original version, L235–240 for the abiotic model and L244 for the whole model). However, to improve clarity, we have now added an overview table of all NN predictors in the Supplementary Materials.

**L216: Are these environmental data continuous time series? If not, what is the effect of patchiness on the outcome?**

We thank the reviewer for this observation. The environmental time series used in our analysis may naturally be patchy as they originate from field-based monitoring. Nevertheless, the 20-year duration of the dataset ensures statistically robust coverage across a wide range of wind

scenarios, light conditions and tidal phases, providing a basis for meaningful analysis. This point was also elaborated in our response to the first point in the general comments.

**L245-247: Chl-a concentration depends mainly on nutrients and grazing. How good are the two chosen proxies?**

We would like to clarify that we used temperature and sunshine duration as an input in the NN model as pragmatic proxies to approximate biological activity, since our dataset did not include more direct biological parameters (e.g. TEP, grazing rates). While these parameters do not capture the whole range of ecological interactions, their purpose in the NN framework is to help distinguish between biologically active and inactive seasons, rather than to capture specific biological mechanisms.

**L247-256: References are missing here. This looks like a description of processes: put in introduction?**

We agree that this descriptive paragraph belongs to the Introduction and have revised it accordingly.

**L252: I am not aware that detritus (is it not always organic?) contributes to flocculation actively. Detritus can be incorporated into flocs by sticky substances such as TEP.**

Following the suggestion in the previous comment, this paragraph has been moved to the Introduction. To more accurately reflect the process, we clarified that the detritus contributes by being incorporated into flocs rather than actively causing flocculation. We also removed "organic" as it was redundant.

*3. Results*

**There is overlap between 3.1 and 3.2 (wind effect is explained in both chapters). I would advise to look only at results without interpretation of the data (keep this for the discussion).**

We appreciate the reviewer's observation regarding potential overlap between Sections 3.1 and 3.2. In the revised manuscript, we have restructured the wind-related content to reduce repetition and clarify the distinction between seasonal patterns (Section 3.1.2) and short-term variability (Section 3.2.2). Specifically, Section 3.1.2 now focuses on the general seasonal co-variation between wind intensity and SPM concentrations, while Section 3.2.2 presents the analysis at finer temporal scales, including the influence of wind averaging ("wind memory") and response lags. We also agree that interpretations should be presented in Discussion rather than in the Results section, and have revised accordingly.

**3.1 Seasonality of SPM concentration**

**L267: Adding a grid for every year would facilitate the reading of the figure**

Thank you for the feedback. In the revised version of the manuscript, to improve the overall readability, we have updated Figure 2 and displayed the time series separately for each year in a grid layout.

**L270-276: This belongs to discussion section (or intro).**

As suggested by the reviewer, we have moved this piece of text to the Introduction and further elaborated in the Discussion.

**L281-284: SPM concentration (and also Chl-a) has a log-normal distribution, taking the arithmetic mean and standard deviation is thus not correct (therefore that the standard deviation is in L284 larger than the mean). A better approach is to use the geometric mean and standard deviation.**

We acknowledge the reviewer's suggestion. While the use of geometric means and standard deviations for SPM and Chl-a concentrations might be a more suitable approach in this context, it may not be familiar to a broader audience and would require additional explanation. Since this section serves only to provide a descriptive overview of the data, we chose to present concentration ranges instead.

**L295: The colors in the figure 3 are confusing. Could you use the same color for SPM and Chl respectively.**

Thank you for the suggestion. Our colour scheme consistently uses blue for the deep station and orange for the shallow station in all figures throughout the manuscript. To maintain consistency and help readers associate each station with a single colour, we chose to differentiate variables (SPM vs. Chl-a) using marker styles and line types rather than colour.

**L316, Fig 4: The large spreading in March suggest that there is a difference between the pre-bloom values (most of them) and the post-bloom ones. Can you check it and maybe adapt the Figure. Further is it useful to keep monthly correlation when the seasons can be split up into winter, spring-bloom, summer, late-summer bloom, autumn?**

We confirm that the data in March indeed reflects a transition between pre-bloom and bloom phases, contributing to higher variability. Since the figure illustrates the data across 20 years and the timing of phytoplankton bloom varies from year to year, it is hard to perfectly delineate the pre-bloom and post-bloom phases. However, we opted to keep the monthly breakdown in Fig. 4 to preserve resolution and highlight intra-seasonal dynamics.

**L299: The difference between both station in terms of correlation is small, delete this statement.**

To avoid overstatement, the sentence has been removed in the revised manuscript.

**L300-303: When resuspension processes are the same, then also deposition ones. For phytoplankton and SPM to settle together means that the phytoplankton is contained in the floc in winter.**

The sentence has been revised to reflect that in winter, phytoplankton is likely associated with flocs, leading to co-settling behavior. We clarified that both resuspension and deposition processes contribute to the observed strong correlation, and that Chl-a may represent a relatively stable component of SPM during this season due to the phytoplankton being contained into settling aggregates.

**L306: $R^2$ is always < 1 per definition**

We agree that this statement was odd and have removed it from the manuscript.

**L306: The sentence is not fully correct. Phytoplankton is part of the SPM, both are retained on the filter. If the correlation weakens, then it means that settling of flocs and phytoplankton differs and thus not all phytoplankton is attached to flocs as in winter. It also means that there is a difference in Chl and SPM concentration over the water column. Near the surface the Chla concentration would be higher, because free phytoplankton will settle slower than biomineral flocs.**

Thank you for the insightful comment. We revised the sentence to clarify that phytoplankton is part of the SPM and co-retained on the filter. The observed weakening of the correlation during biologically active periods likely reflects the decoupling of active phytoplankton from flocs, resulting in different settling behaviors. While sinking rates are higher for the flocs, freely suspended phytoplankton may remain concentrated in surface layers, leading to vertical separation and reduced correlation between Chl-a and total SPM.

**L310: Which broader variables?**

To clarify this point, in the revised manuscript, we added a table to explicitly list all the input parameters included in the Neural Network, both for the winter (only abiotic) model and all seasons (abiotic + light availability, temperature).

**L324: Did you use all wind data? I would suggest to use only the wind data at the time of the samples and to compare this curve with the SPM concentration, as I suspect that the sampling has a biais towards good weather.**

Wind stress causes resuspension as an immediate response, but we also need to account for wind conditions several hours prior to sampling. Because resuspended material can remain in suspension and be redistributed by tidal currents for several hours, wind conditions preceding the sampling may also affect observed concentrations. Accordingly, for Fig.5, we did use the wind data at the time of sampling  (wind dataset has an hourly resolution, and for our analysis, we take the wind data at the same hour as samples are taken). In the subsequent statistical analysis, when we consider the averaged wind speed, we average the hourly resolved data for the past n hours.

**L339-343: I see that the correlation is overall very weak, although often significant. This means that the instantaneous wind speed does not explain a large portion of the variability in SPM concentration. Figure 6 seems not necessary for me.**

We partially agree with the comment. Although the correlations are not particularly strong, considering the non-linear processes in play, they are also not negligible. The correlations are also substantial in winter as opposed to summer, which is the mechanism we intended to highlight. Moreover, within winter correlations, we see that the shallow station consistently responds to the wind when we correlate it at the time of sampling. This distinction leads us to a deeper understanding of the differences between shallow and deep stations.

**L351: 120 hours, is this prior to the sampling date?**

This is the average wind speed for 120 hours before the sampling time.

**L353-356: Still the correlation is low, meaning that wind speed can only partially explain some of the observed variability in SPM concentration.**

We recognize that "stronger" may be misleading and have revised the sentence to reflect a more accurate interpretation of the correlation strength. However, we would like to stress that the correlations and their statistical significance improve for the deep station when we apply the average of the past 120-hour wind speed. Both were compared to the wind at the time of sampling for the deep station, *and* compared to the 120-hour averaged wind speed correlated for the shallow station.

**L357: 'crucial role': this phrasing is exaggerated, delete 'crucial'**

We revised the sentence to avoid overstatement.

**L359: What is the role of 'indirect resuspension', which I would consider as wave induced resuspensions. Do you have information on waves? I would expect that the role of waves in resuspending sediments is higher than of wind. Wind will result in changes of the advective transport of particles and waves in resuspension.**

Wind stress determines both the 'direct' resuspension caused by induced shear in the water column and the 'indirect resuspension' resulting from wave breaking. While direct wave measurements were not available for this study, their influence is implicitly represented in the wind and tide datasets used in the NN framework. Because the NN captures nonlinear interactions, both direct and indirect resuspension mechanisms are taken into account, although their individual contributions cannot be disentangled explicitly.

**3.2 Resuspension and Time scales of Inner Basin Transport**

**3.2.1: Is not needed**

We acknowledge that this subsection is more of supportive nature and might dilute the focus of the study. We moved it to Supplementary Materials.

**L373-381 and Figure 8: I only see marginal differences between low and high tide. I would suggest to remove the Figure and the text. You mention sampling depth, see my comment for L187-188.**

In such a tidal basin with shallow bathymetry, the depth of the water column, which is reflected by the sea surface height (SSH) influences the concentrations of a sampled material. However, as it is not the main focus of the study, we decided to move this part to the Supplementary Material.

**L383: The difference in current velocities between ebb and flood are a better indicator for SPM concentration during sampling than its (non) relation with SSH.**

The phase of the tidal cycle determines the velocity field. Due to the strong role of non-linear processes in the domain, ebb and flood are not equally strong in terms of maximum and mean velocities. This asymmetry is evident in the figure below, where SPM concentrations are shown

together with the SSH gradient (derivative is calculated using forward scheme, time step is ~5 min) and velocity asymmetry:

[Figure]

Figure: Relationship between suspended particulate matter (SPM) concentrations and tidal phase, represented by the gradient of reconstructed sea surface height (SSH), at the deep and shallow stations (left, middle). Ebb and flood phases show asymmetric distributions, with highest SPM concentrations occurring during flood. The spatial pattern of mean transport velocity asymmetry (ebb-to-flood ratio) in the Sylt-Rømø Bight (right) further illustrates that ebb and flood currents are not equally strong, reflecting non-linear tidal dynamics in the basin.

We also refer to our earlier work (Fofonova et al., 2019; Konyssova et al., 2025) for a more detailed analysis of tidal asymmetry in this system, and we have expanded the Discussion section to reflect on this aspect.

**L400-405: In Figure 7 you used a 120h period for both stations, although 12h would have been better for the shallow station? Again, the R² is low, so that the explanatory variability of wind on SPM concentration is small. Figure 10 is not necessary.**

Figure 10 has been placed in the Supplementary Materials, as we agree that the value added is very limited.

However, the reason we chose to show 120 hours originates from the results we observe in Figure 9, where the peak correlation of wind speed and SPM concentrations at the deep station is observed in around 120 hours.

**L419-420: Seems logic to me that a storm in summer has a similar effect (increases resuspension) as a storm in winter. Delete 'even' in L419**

Revised accordingly.

**L426: Although there is a link between SPM and a passive tracer, it is not the same. How representative are the results shown in Figure 11. Why not add SPM characteristics to the passive tracers (critical shear stress for erosion and settling velocity)?**

Thank you for raising this point. We agree that passive tracers do not fully reproduce the behavior of suspended particulate matter, especially when it comes to settling, flocculation, or resuspension thresholds. Unfortunately, we lack the sediment-specific parameters needed to support such sediment simulations. As outlined in our response to point 2 in the general

comments, the purpose of this simplified setup was not to accurately model SPM fate, but to illustrate hydrodynamic connectivity under tidal forcing and to identify dominant source areas and transport pathways to the stations. We have clarified this more explicitly in the revised manuscript.

**L430: Can you explain what you mean by 'stochastic fluctuations'?**

By "stochastic fluctuations", we refer to the irregular, short-term variability in wind-driven transport dynamics that can enhance dispersion but do not significantly contribute to basin-wide transport, which is predominantly driven by tides.

**L431: 80% of the velocity variability is due to tide, do you mean by 'relatively minor (wind forcing)' the remaining 20%? You should be more specific.**

We appreciate the opportunity to clarify this point. The statement regarding "relatively minor" refers not to the 20% in velocity variability presented in Fofonova et al. (2019), but to results from our previous Lagrangian transport analysis (Konyssova et al., 2025). That study showed that adding summer wind forcing increased particle dispersion but did not alter the dominant transport pathways, which remained primarily driven by non-linear tidal dynamics. Meanwhile, the ~20% residual in velocity variability reported by Fofonova et al. (2019) includes not only wind-driven effects but also baroclinic circulation.

**L435-439: Has been mentioned in the Method section**

As suggested, we removed this piece from the Results to avoid redundancy.

**L439-443: There are differences, but both are also very similar.**

We agree that both stations share common transport routes, which is expected given their proximity and location relative to the tidal inlet and the main tidal channel (Lister Ley). However, the probability distribution reveals distinct differences in dominant source regions. While the shallow station shows a clear concentration of tracer origins in the Königshafen embayment, the deep station is fed by a broader and more dispersed area, with relatively low contributions from Königshafen. This spatial variation is meaningful and reflects differences in connectivity patterns that are important to highlight.

**L457: See my previous comment, I am not convinced that passive tracer give the same result as 'active' tracers (such as SPM).**

We agree that passive tracer simulations cannot fully represent the transport behavior of active particulate matter such as SPM. The figures are intended to illustrate the hydrodynamic connectivity between the stations and the surrounding basin under tidal forcing, helping to identify likely source regions and transport pathways. We acknowledge that the phrase "comprehensive picture of SPM transport" may have overstated the scope of this experiment, and we have revised the wording accordingly to clarify the intent and avoid overstatement.

**3.3 Neural Network**

**L492: Salinity as a proxy for baroclinic conditions has not been explained. How do barclinic conditions affect SPM concentrations?**

Salinity was included as a proxy for baroclinic effects, particularly the density gradients that locally drive the circulation in the area (Burchard et al., 2008).

**L497-501: The effect of temperature on flocculation in marine areas is small, and the effect of salinity on flocculation is important at the limit of the sea water intrusion in estuaries (so at low salinities).**

We acknowledge the reviewer's comment. We consider that the influence of temperature and salinity on flocculation varies depending on site-specific dynamics. In our study region, salinity variability is limited due to minimal freshwater input, so its effect on flocculation is expected to be minor. Temperature, on the other hand, while not a strong flocculation driver on its own, was considered due to its indirect influence on biological processes (e.g., phytoplankton growth, bacterial metabolism, EPS production). We have revised the text to reflect these distinctions more accurately and avoid overstating the direct role of these parameters.

**L514: How did you obtain that wind explains 40% of the decrease?**

The ~40% refers to the difference in median SPM concentrations between winter and summer *predicted* by the NN trained only on physical drivers. Specifically, the NN-predicted values in summer remain ~40% higher than the observed SPM concentrations, indicating that wind conditions alone cannot fully explain the seasonal drop.

**L518-520: How did you define the seasons? As mentioned before, spring (March) may contain some typical winter conditions (pre-bloom) and autumn maybe some typical late summer conditions (summer bloom). Both may blur your results. To explain it by wind conditions is therefore not convincing.**

We acknowledge the reviewer's point and agree that biological and physical transitions do not necessarily align with calendar months. We used seasonal boundaries based on observed seasonal cycles in Chl-a concentrations at the study site rather than using calendar-based or astronomical definitions. Specifically, we used the following delineations:
-   Winter: November 20 – February 19 (low biological activity, low Chl-a)
-   Spring: February 20 – May 31 (phytoplankton bloom initiation and peak)
-   Summer: June 1 – September 19 (post-bloom conditions, high light, reduced Chl-a)
-   Autumn: September 20 – November 20 (transitional period).

The overestimations discussed in the NN regression plots are therefore interpreted in the context of these adjusted seasonal windows. This has been missed in the original version, and has been explicitly clarified in the revised text.

**L538-542: Is also included in tch 3.3.1**

We revised the text and removed the repeated information.

*4. Discussion*

**L549-550: Biological processes are only introduced indirectly. There is no evidence of phytoplankton biomass or grazing in your data set.**

We acknowledge that our dataset does not include direct measurements of phytoplankton biomass, grazing, or exopolymers. As mentioned earlier, chlorophyll-a is used here as a general proxy for phytoplankton presence and associated biological processes. Since this is a proxy rather than direct evidence, we cannot resolve individual mechanisms such as flocculation or grazing due to the absence of more detailed biological data. Nevertheless, it provides a basis for exploring the relative contribution of biologically mediated processes to SPM concentration variability. We have elaborated on this in our response to point 4 in the general comments.

**4.1 Biological interactions**

**L562-564: Chl is not a good proxy for biological mediated flocculation. The latter is driven by the occurrence of exopolymers that have been excreted by the phytoplankton.**

We agree that biologically mediated flocculation is driven by exopolymers such as TEP and EPS rather than directly by phytoplankton biomass. However, given the absence of direct measurements of these substances or phytoplankton species composition, we used Chl-a as a general proxy for biological activity, including but not limited to flocculation.

**L572-575: The decoupling between SPM and Chl-a from spring onward, does not explains the formation of larger flocs with higher settling velocity. Further, the formation of larger flocs occurs already at the start of the phytoplankton bloom and not only at the end.**

Thank you for highlighting this point. The cited studies show that flocculation intensifies both at the onset of the bloom and during its decline. Our revised sentence reflects this timing, making clear that larger and more rapidly settling flocs form early in the bloom and continue afterwards, helping explain the observed decoupling between Chl-a and SPM concentrations.

**L575-577: Can you give a number that shows how big the influence of temperature is.**

Temperature affects SPM both directly through physical processes and indirectly by modulating biological activity. To evaluate its role, we tested the full-year model with abiotic parameters only excluding temperature and compared it against a model where temperature was included.  The improvement in correlation was modest in winter (R = 0.66 to 0.7) but substantial in other seasons:  from 0.61 to 0.78 in spring, from 0.38 to 0.73 in summer, and 0.56 to 0.76 in autumn  (table below; also included in the Supplementary Materials). Additionally, a model with two biological proxies (temperature and sunshine hours) was added in the last column, which shows a further improved correlation. The results indicate that the combined effect of temperature and light is minor in winter but critical for capturing SPM variability during biologically active seasons.

Table: Mean, median values and correlation (R) of SPM concentrations, [mg/L], from observations versus predictions by NN trained on all abiotic dataset (no temperature), all abiotic dataset + temperature,  and all abiotic dataset + temperature & light for the Deep Station.

| seasons | | observed | NN: all seasons abiotic (no temperature) | NN: all seasons abiotic + temperature | NN: all seasons abiotic + temperature, light |
|---|---|---|---|---|---|
| winter | mean median, [mg/L] | 27.6 25.9 | 19.2 17.2 | 26.3 23.7 | 25.6 23.6 |
| | | | R = 0.66 | R = 0.7 | R = 0.73 |
| spring | mean median, [mg/L] | 16.5 13.3 | 14.8 13.17 | 17 14.2 | 16.6 14.3 |
| | | | R = 0.61 | R =0.78 | R =0.82 |
| summer | mean median, [mg/L] | 6.2 4.2 | 7.9 7.1 | 7.1 5.8 | 7.1 5.8 |
| | | | R =0.38 | R =0.65 | R =0.73 |
| autumn | mean median, [mg/L] | 11.6 8.5 | 12.6 10.8 | 13.3 11.2 | 11.9 9.3 |
| | | | R = 0.56 | R =0.76 | R =0.78 |

**L585-588: Filter feeders will remove SPM from the system by producing fecal pellets. The POC fraction in the SPM depends on the SPM concentration (see Schartau et al., 2019).**

The revised paragraph specifies, with reference to Schartau et al. (2019), that the POC fraction in SPM varies with total SPM concentration, which influences the efficiency and variability of organic matter removal.

**L588-595: What is the relevance for your study? They remain speculative when you do not have data about it.**

The relevance of this paragraph lies in situating our findings within the broader ecological context of the Sylt-Rømø Bight. Therefore, we are briefly discussing the potential contribution of bentho-pelagic coupling, including from introduced species, on the variability of SPM concentrations. We have revised the paragraph to clarify that this point is raised as a potential direction for future work rather than a confirmed mechanism found in this study.

**4.2 Wind & Tidal Control**

**This chapter is a summary of the results and not a discussion.**

We revised Section 4.2 to shift the focus from summarizing results toward a more interpretative discussion. Descriptive elements have been reduced or condensed, and we now highlight how the observed timescales and station-specific responses relate to sediment characteristics, transport pathways, and broader findings from the literature (e.g., de Jonge and van Beusekom,

1995; Friedrichs and Perry, 2001; Winterwerp, 2001). These changes aim to clarify the implications of the wind and tidal forcing results and situate our findings within the broader context of SPM dynamics in shallow tidal systems.

**4.3 Neural Network**

**This chapter is a summary of the results and not a discussion.**

Following the reviewer's suggestion, we revised the section to reduce repetition of results and focus more clearly on the implications of the observed patterns. The revised text emphasizes how the neural network results support and extend the process-based interpretation of seasonal shifts in dominant SPM drivers. We also highlight the added value of the NN approach in capturing non-linear interactions and complementing statistical methods, especially in complex, biologically influenced periods where data limitations exist.

**5. Conclusion**

**L712-713: The correlation between wind speed and SPM concentration was never strong.**

Thank you for pointing this out. We acknowledge that the phrase could be misleading given the complex and non-linear nature of the mechanisms influencing SPM concentration variability. To avoid overstating the correlations and to reduce redundancy with the preceding sentence, we have revised the sentence to emphasize the role of wind stress in shaping short-term variability.

....................................................................................................................................................

References:

Burchard, H., Bolding, K., Rippeth, T. P., Stips, A., Simpson, J. H., and Sündermann, J.: Microstructure of turbulence in the northern North Sea: a comparative study of observations and model simulations, Journal of Sea Research, 47, 223–238, https://doi.org/10.1016/S1385-1101(02)00126-0, 2002.

Burchard, H., Flöser, G., Staneva, J. V., Badewien, T. H., and Riethmüller, R.: Impact of Density Gradients on Net Sediment Transport into the Wadden Sea, Journal of Physical Oceanography, 38, 566–587, https://doi.org/10.1175/2007JPO3796.1, 2008.

Cloern, J. E.: The relative importance of light and nutrient limitation of phytoplankton growth: a simple index of coastal ecosystem sensitivity to nutrient enrichment, Aquatic Ecology, 33, 3–15, https://doi.org/10.1023/A:1009952125558, 1999.

Fettweis, M., Monbaliu, J., Baeye, M., Nechad, B., and Van Den Eynde, D.: Weather and climate induced spatial variability of surface suspended particulate matter concentration in the North Sea and the English Channel, Methods in Oceanography, 3–4, 25–39, https://doi.org/10.1016/j.mio.2012.11.001, 2012.

Fofonova, V., Androsov, A., Sander, L., Kuznetsov, I., Amorim, F., Hass, H. C., and Wiltshire, K. H.: Non-linear aspects of the tidal dynamics in the Sylt-Rømø Bight, south-eastern North Sea, Ocean Sci., 15, 1761–1782, https://doi.org/10.5194/os-15-1761-2019, 2019.

Friedrichs, C. and Perry, J.: Tidal salt marsh morphodynamics: a synthesis, Journal of Coastal Research, 27, 7–37, 2001.

de Jonge, V. N. and van Beusekom, J. E. E.: Wind- and tide-induced resuspension of sediment and microphytobenthos from tidal flats in the Ems estuary, Limnol. Oceanogr., 40, 776–778, https://doi.org/10.4319/lo.1995.40.4.0776, 1995.

Konyssova, G., Sidorenko, V., Androsov, A., Sander, L., Danilov, S., Rubinetti, S., Burchard, H., Winter, C., and Wiltshire, K. H.: Changes in tidal dynamics in response to sea level rise in the Sylt-Rømø Bight (Wadden Sea), Ocean Dynamics, 75, 43, https://doi.org/10.1007/s10236-025-01688-1, 2025.

Rick, J. J., van Beusekom, J., Romanova, T., and Wiltshire, K. H.: Long-term physical and hydrochemical measurements at Sylt Roads LTER (1973-2013), Wadden Sea, North Sea, links to data sets, https://doi.org/10.1594/PANGAEA.150032, 2017.

Rick, J. J., Scharfe, M., Romanova, T., Van Beusekom, J. E. E., Asmus, R., Asmus, H., Mielck, F., Kamp, A., Sieger, R., and Wiltshire, K. H.: An evaluation of long-term physical and hydrochemical measurements at the Sylt Roads Marine Observatory (1973–2019), Wadden Sea, North Sea, Earth Syst. Sci. Data, 15, 1037–1057, https://doi.org/10.5194/essd-15-1037-2023, 2023.

Schartau, M., Riethmüller, R., Flöser, G., Van Beusekom, J. E. E., Krasemann, H., Hofmeister, R., and Wirtz, K.: On the separation between inorganic and organic fractions of suspended matter in a marine coastal environment, Progress in Oceanography, 171, 231–250, https://doi.org/10.1016/j.pocean.2018.12.011, 2019.

Stanev, E. V., Dobrynin, M., Pleskachevsky, A., Grayek, S., and Günther, H.: Bed shear stress in the southern North Sea as an important driver for suspended sediment dynamics, Ocean Dynamics, 59, 183–194, https://doi.org/10.1007/s10236-008-0171-4, 2009.

UNESCO (1966) Determination of Photosynthetic Pigments in Sea Waters. Monographs on oceanographic methodology, Paris, 66 p.

---

## Referee Report (RR1)

**Second round review** of the manuscript *Wind and Phytoplankton Dynamics Driev Seasonal and Short-Term Variability of Suspended Matter in a Tidal Basin*, by Konyssova et al, submitted to **Biogeosciences** (egusphere-2025-2135).

**Review overview**

The authors have revised their manuscript thoroughly text-wise and I am happy to see that they are much more nuanced now with their statements and conclusions. A few minor remarks remain which are listed below.

**Recommendation**

Minor revision, recommending publication.

**Detailed Comments**

1. Line 54-55: I assume the tides will also cause erosion under severe weather conditions, except that waves will cause larger erosion in those circumstances.

2. Line 78-79: I miss references here for the statement about the effect of zooplankton grazing on organic aggregates. And it makes me wonder what the process would be behind it.

3. Line 115-116: needs rephrasing. At the very least something like "The basin … embayment called Konigshafen, which has an average depth of ~2m and encompasses large areas which become exposed at low tide", or more simply "encompasses large intertidal areas".

4. Line 160 "displayed in Fig. 1 " gives the impression that the results will be shown there, rather than just the location of the stations. Something like "(see Fig. 1)" would be better.

5. Line 227: not every reader may be aware of what LTER stands for, and it is not explained anywhere in the text currently.

6. Line 236-238: I do not agree with this. Discrepancies between numerical models and observational evidence (which is what is meant here I assume) can be due to many things, including the spatial and temporal resolution of the model, the spatial and temporal resolution of the observations, indirect observational evidence, lack of processes within the model, lack of accurate forcing data like initial conditions or boundary conditions or the temporal resolution of the applied meteorology. I suggest reading Skogen et al (2021) for a more nuanced view. But my interpretation here was that a full numerical model would be costly to run and add little for a first quantification of the different drivers. It would, however, have added better process understanding than a NN model can provide, but would be unable to fully capture the short-term bursts of wind that cause resuspension. Hence I support the choice for the NN model.

7. Line 364: according to the graphs the winter values of Dec-Feb (0.30, 0.44, 0.31) are not much different from those of spring (Mar-May: 0.40, 0.50, 0.10) at the shallow station, average wise. So why is winter listed as having the highest correlation?

8. Line 429-430: or highlighting the different source regions? Possibly connected to differences in grain sizes?

9. Fig. 11: the right side graph is of very poor quality, both digital and in print.

10. Line 489-490: you cannot prove a negative. I would say that the regression coefficient dropping to near zero only indicates other drivers, and that your subsequent analysis for all seasons indicates that it is a biological one.

11. Line 523: "NN related fundings", did the model receive payment for its work?

12. Line 588: I don't understand the use of "as soon as" here, I assume you simple mean "as"?

13. Line 695: "from the intertidal and shallow *areas*"

**References**

Skogen, M.D., Ji, R., Akimova, A., Daewel, U., Hansen, C., Hjollo, S.S., van Leeuwen, S.M., Maar, M., Macias, D., Mousing, E.A., Almroth-Rosell, E., Sailley, S.F., Spence, M.A., Troost, T., van de Wolfshaar, K. (2021) *Disclosing the truth: are models better than observations?*, Marine Ecology Progress Series, DOI: 10.3354/meps13574

---

## Author Response (AR2)

**Second round review** of the manuscript *Wind and Phytoplankton Dynamics Drive Seasonal and Short-Term Variability of Suspended Matter in a Tidal Basin* , by Konyssova et al, submitted to **Biogeosciences** (egusphere-2025-2135).

We would like to sincerely thank the Editor and Reviewers for their thoughtful and constructive feedback. We greatly appreciate the time and effort dedicated to evaluating our revised manuscript. The comments have helped us improve the clarity, balance, and interpretation of our study.

Below, we provide a detailed point-by-point response to the comments and describe the corresponding revisions made in the manuscript. Reviewer comments are shown in **bold**, and our responses follow in regular font.

………………………………………………………………………………………………………………

Response to Reviewer #1

**The authors have taken my comments into consideration and I agree with most of their answers. However, I am not convinced by the conclusions of the manuscript, especially the high importance given to wind effects.**

**To my opinion the authors underestimate the biological effects (TEP production) during spring and summer and its influence on flocculation and seabed stability (check the literature, where many recent papers highlight the importance of seasonal biological effects on flocculation and SPM concentration). Biological effects explain a large part of the summer decrease in SPM concentration (probably more than the outcome of your analysis). In contrast, in winter the sticky TEPs are absent and as a consequence flocs are smaller than in summer and the critical shear stress for erosion is lower. The absence of TEP in winter explains to a large part the higher SPM concentration to my opinion. Wind increases resuspension, but remains less important than the seasonal biological cycle, because the frequency and duration of storm events are limited.**

We fully agree with the reviewer's comment on the role of biological effects in spring and summer. In the previous version of the manuscript, we emphasized that biological processes become increasingly dominant with the onset and progression of the phytoplankton bloom (L536-541). We also acknowledge that the data available for the NN analysis may not have been sufficient to fully capture the biological feedbacks, since more drivers are at play than those we included as proxies in our model (L541-544). Nevertheless, we agree that the original conclusions may have placed relatively strong emphasis on wind forcing, and we appreciate the reviewer's more nuanced perspective. In response, we have adjusted the tone of the Conclusions to better reflect that physical

and biological mechanisms operate on different temporal scales. We now clarify that while wind and tidal forcing remain essential in shaping high-frequency variability, particularly in winter and autumn when biological activity is low, the biologically mediated processes increasingly dominate during spring and summer by suppressing SPM concentrations to lower levels despite similar wind forcing.

**What is the role of tidal forces in the seasonal SPM concentration cycle? This has not been investigated or discussed and could be studied by comparing periods in summer and winter with negligible winds. Such a comparison could provide an estimate of biological effects and could strengthen or weakens your conclusion on seasonal wind effects.**

Regarding the role of tidal forces in the seasonal SPM concentration cycle, we do not observe notable seasonality in the tidal dynamics. Although tide itself does not change throughout the year, the effectiveness of tidal currents in mobilizing and transporting particles likely varies seasonally because the properties of the sediment and the level of biological stabilization change over time. Therefore, in addition to the other factors with clear seasonal signal, we also included tidal data in the NN analysis to account for its overall influence. However, we acknowledge that we did not specifically quantify the contribution of the tides to seasonal variability of SPM concentrations. We thank the reviewer for raising this point as it adds an interesting point of discussion on the role of tides.

**It is actually indirectly shown in Figures 7 and 8: the range of wind speed values is similar during the year, however, there is almost no correlation between SPM concentration and wind speed during the summer months. There is further something contradictory when comparing Figures 7 and 8 with Figure 5. Figure 5 suggests a strong correlation between wind speed and SPM concentration, which is in contrast with Figures 7 and 8, which show a very low correlation. To my opinion and based on literature on biological effects, Figure 5 is misleading as it overestimate the wind influence and does not take biological effects into account. Temperature as a proxy of biological effects would show another correlation. Is the frequency of wind/storm periods enough to explain the correlation shown in Figure 5 between SPM concentration and wind speed? You would need almost every 5 days a storm in winter to explain it, I guess.**

The reviewer is also right that the Fig. 5 does not take biological effects into account, because we wanted to analyze the abiotic and biological effects in seperate. The figure was also not intended to represent a direct correlation between wind speed and SPM concentration but rather to illustrate that they both follow a similar seasonal pattern through the year, with high values during winter and lower values in summer.

The Figures 7 and 9, however, provide the monthly correlations which look into the variability of wind speed and SPM concentrations within respective months. We agree that the monthly correlation coefficients between wind speed and SPM concentrations are not particularly strong and we have also not stated that the wind plays dominant role across all seasons, but it is the main driver when the biological activity is minimal. Beginning with and following the phytoplankton bloom in early spring, the biological effects have increasingly larger influence.

**In an answer to a comment of mine, you have ascribed the low correlation between SPM concentration and wind speed to the inherent complexity of the system, does this means that the stochasticity of the system is more important than wind effects or does this complexity covers to a larger part the effect of tides?**

We thank the reviewer for raising this point. Indeed, the correlation coefficients between SPM concentrations and wind speed are not particularly strong, as winds alone cannot explain the full variability of the SPM concentrations across all seasons. By referring to the inherent complexity of the system, we meant that the observed SPM variability results from the interaction of multiple driving mechanisms including tides, wind, and biologically modulated processes, each operating at different temporal and spatial scales. So the relatively low correlations likely reflect the presence of non-linear interplay among these drivers rather than random stochastic behavior. To clarify this point, we have revised the Discussion section to elaborate that the system complexity arises from the superposition of physical and biological processes, which vary seasonally in their relative importance.

……………………………………………………………………………………………………………

Response to Reviewer #2

**Review overview**

**The authors have revised their manuscript thoroughly text-wise and I am happy to see that they are much more nuanced now with their statements and conclusions. A few minor remarks remain which are listed below.**

**Detailed Comments**

1. **Line 54-55: I assume the tides will also cause erosion under severe weather conditions, except that waves will cause larger erosion in those circumstances.**

   We agree that the original phrasing could be misleading, as it may imply that tidal erosion occurs only under calm conditions. The revised sentence now clarifies that tidal currents maintain a baseline shear stress and govern sediment transport

under all weather conditions, while during severe weather, waves cause more intense erosion superimposed with the tidal flow.

2.  **Line 78-79: I miss references here for the statement about the effect of zooplankton grazing on organic aggregates. And it makes me wonder what the process would be behind it.**

    We thank the reviewer for noting the need for clarification and appropriate references. The revised text now expands on the underlying processes through which zooplankton grazing influences SPM concentration and composition. The effect occurs both directly through the consumption and fragmentation of phytoplankton aggregates and the production of fecal pellets, and indirectly through the release of organic matter that enhances microbial and TEP formation (e.g., Passow, 2002; Toullec et al., 2019; Turner, 2002). These processes collectively alter particle size, cohesion, and settling behavior, which potentially leads to either enhanced aggregation and sedimentation or increased recycling within the water column.

3.  **Line 115-116: needs rephrasing. At the very least something like "The basin … embayment called Konigshafen, which has an average depth of ~2m and encompasses large areas which become exposed at low tide", or more simply "encompasses large intertidal areas".**

    We rephrased the sentence as suggested.

4.  **Line 160 "displayed in Fig. 1 " gives the impression that the results will be shown there, rather than just the location of the stations. Something like "(see Fig. 1)" would be better.**

    Revised accordingly.

5.  **Line 227: not every reader may be aware of what LTER stands for, and it is not explained anywhere in the text currently.**

    Expanded to include the full term *Long-Term Ecological Research (LTER)* at its first mention (L98).

6.  **Line 236-238: I do not agree with this. Discrepancies between numerical models and observational evidence (which is what is meant here I assume) can be due to many things, including the spatial and temporal resolution of the model, the spatial and temporal resolution of the observations, indirect observational evidence, lack of processes within the model, lack of accurate forcing data like initial conditions or boundary conditions or the temporal resolution of the applied meteorology. I suggest reading Skogen et al (2021) for a more nuanced view. But my interpretation here was that a**

**full numerical model would be costly to run and add little for a first quantification of the different drivers. It would, however, have added better process understanding than a NN model can provide, but would be unable to fully capture the short-term bursts of wind that cause resuspension. Hence I support the choice for the NN model.**

We acknowledge the reviewer's comment that the discrepancies between modelled and observational can be attributed to various factors, including the representation of physical and biological processes in models, boundary conditions, and the spatial and temporal resolution of both modelled and observational data. In the revised version, we have incorporated the suggested reference (Skogen et al., 2021) and expanded our reasoning to include more nuanced view. We also thank the reviewer for supporting our choice to use the NN approach as an efficient way to capture complex, non-linear dependencies among the measured variables, while recognizing that it does not provide process understanding.

7. **Line 364: according to the graphs the winter values of Dec-Feb (0.30, 0.44, 0.31) are not much different from those of spring (Mar-May: 0.40, 0.50, 0.10) at the shallow station, average wise. So why is winter listed as having the highest correlation?**

We agree with the reviewer's observation. The revised phrasing now specifies that the correlations are comparably strong in both winter and early spring. This pattern likely reflects a seasonal transition in which biological effects start to have an increasing influence, yet wind-driven resuspension still significantly contributes to short-term variability of SPM concentrations.

8. **Line 429-430: or highlighting the different source regions? Possibly connected to differences in grain sizes?**

We appreciate this suggestion from the reviewer. In fact, the difference between the median transit times at the deep and at the shallow stations does highlight the different source zones and associated and transport pathways feeding each station. While our passive tracer simulations do not explicitly resolve grain-size classes, we agree that the different in sediment characteristics would influence the transport timing. In the revised manuscript, we mention these in the text as an additional explanation.

9. **Fig. 11: the right side graph is of very poor quality, both digital and in print.**

We thank the reviewer for noting the issue with the figure quality. The figure has been replaced with a higher-resolution version.

10. **Line 489-490: you cannot prove a negative. I would say that the regression coefficient dropping to near zero only indicates other drivers, and that your subsequent analysis for all seasons indicates that it is a biological one.**

    This is a valid point, and we thank the reviewer for this correction. A near-zero coefficient indeed does not prove the absence of wind influence, but rather reflects that other drivers become dominant, which is then demonstrated in the following section. We have rephrased the sentence accordingly.

11. **Line 523: "NN related fundings", did the model receive payment for its work?**

    Thank you for noticing the typo, we corrected it to "NN-related findings."

12. **Line 588: I don't understand the use of "as soon as" here, I assume you simple mean "as"?**

    Replaced "as soon as" with "as" for clarity.

13. **Line 695: "from the intertidal and shallow *areas*"**

    Revised the phrase accordingly.

**References**

Passow, U.: Transparent exopolymer particles (TEP) in aquatic environments, Progress in Oceanography, 55, 287–333, https://doi.org/10.1016/S0079-6611(02)00138-6, 2002.

Skogen, M., Ji, R., Akimova, A., Daewel, U., Hansen, C., Hjøllo, S., Van Leeuwen, S., Maar, M., Macias, D., Mousing, E., Almroth-Rosell, E., Sailley, S., Spence, M., Troost, T., and Van De Wolfshaar, K.: Disclosing the truth: Are models better than observations?, Mar. Ecol. Prog. Ser., 680, 7–13, https://doi.org/10.3354/meps13574, 2021.

Toullec, J., Vincent, D., Frohn, L., Miner, P., Le Goff, M., Devesa, J., and Moriceau, B.: Copepod Grazing Influences Diatom Aggregation and Particle Dynamics, Front. Mar. Sci., 6, 751, https://doi.org/10.3389/fmars.2019.00751, 2019.

Turner, J.: Zooplankton fecal pellets, marine snow and sinking phytoplankton blooms, Aquat. Microb. Ecol., 27, 57–102, https://doi.org/10.3354/ame027057, 2002.